# SpotClean adjusts for spot swapping in spatial transcriptomics data

Zijian Ni [1,6], Aman Prasad[2,6], Shuyang Chen[1], Richard B. Halberg[3,4], Lisa M. Arkin[2], Beth A. Drolet[2], Michael A. Newton[1,5] & Christina Kendziorski[5 ✉]

Spatial transcriptomics is a powerful and widely used approach for profiling the gene expression landscape across a tissue with emerging applications in molecular medicine and tumor diagnostics. Recent spatial transcriptomics experiments utilize slides containing thousands of spots with spot-specific barcodes that bind RNA. Ideally, unique molecular identifiers (UMIs) at a spot measure spot-specific expression, but this is often not the case in practice due to bleed from nearby spots, an artifact we refer to as spot swapping. To improve the power and precision of downstream analyses in spatial transcriptomics experiments, we propose SpotClean, a probabilistic model that adjusts for spot swapping to provide more accurate estimates of gene-specific UMI counts. SpotClean provides substantial improvements in marker gene analyses and in clustering, especially when tissue regions are not easily separated. As demonstrated in multiple studies of cancer, SpotClean improves tumor versus normal tissue delineation and improves tumor burden estimation thus increasing the potential for clinical and diagnostic applications of spatial transcriptomics technologies.

[1] Department of Statistics, University of Wisconsin-Madison, Madison, WI, USA. [2] Department of Dermatology, University of Wisconsin-Madison, Madison, WI, USA. [3] Department of Medicine, University of Wisconsin-Madison, Madison, WI, USA. [4] Department of Oncology, University of Wisconsin-Madison, Madison, WI, USA. [5] Department of Biostatistics and Medical Informatics, University of Wisconsin-Madison, Madison, WI, USA. [6] These authors contributed equally: Zijian Ni, Aman Prasad. ✉email: kendzior@biostat.wisc.edu

Spatial transcriptomics (ST) is a powerful and widely used approach for profiling genome-wide gene expression across a tissue[1,2]. In a typical ST experiment, fresh-frozen (or FFPE) tissue is sectioned and placed onto a slide containing spots, with each spot containing millions of capture oligonucleotides with spatial barcodes unique to that spot. The tissue is imaged, typically via Hematoxylin and Eosin (H&E) staining. Following imaging, the tissue is permeabilized to release RNA which then binds to the capture oligonucleotides, generating a cDNA library consisting of transcripts bound by barcodes that preserve spatial information. Data from an ST experiment consists of the tissue image coupled with RNA sequencing data collected from each spot. A first step in processing ST data is tissue detection, where spots on the slide containing tissue are distinguished from background spots without tissue. Unique molecular identifier (UMI) counts at each spot containing tissue are then used in downstream analyses (Supplementary Fig. 1).

Ideally, a gene-specific UMI at a given spot would represent expression of that gene at that spot. This is not the case in practice. As we demonstrate here, messenger RNAs bleed between and among nearby spots causing substantial contamination of UMI counts, an artifact we refer to as spot swapping. Spot swapping is related to, but distinct from, previously defined sources of contamination which have been widely recognized over the past decade in next-generation sequencing studies[3]. Specifically, improvements in sequencing technologies have greatly increased the speed and scale at which data can be obtained, but the advantages rely on multiplexing where indexes (or barcodes) are attached to each RNA (or DNA) fragment in a sample prior to pooling so that sample-specific transcripts can be identified in the sequenced pool. In spite of the major advantages in reduced cost and increased efficiency, a disadvantage is that indexes from one sample may attach to transcripts from another, an error referred to as index hopping or barcode swapping. While present in most datasets[3–6], good statistical methods are in place to adjust for this type of contamination[4–6]. A second type of contamination is specific to single-cell RNA sequencing (scRNA-seq) experiments, where ambient RNA is sequenced along with RNAs from an individual cell. As with index hopping, robust statistical methods are in place to adjust for ambient RNA contamination in droplet-based scRNA-seq experiments[7,8].

While existing methods can be used to remove some sources of contamination present in an ST experiment, they are insufficient to adjust for effects due to spot swapping as they do not accommodate the spatial nature of the contamination (for further discussion, see Supplementary Section S1). In barcode swapping, the swapping takes place in the pooled cDNA library, and so a barcode from one sample has an equal chance of binding reads from any another sample. In spot swapping, barcodes from one spot are much more likely to bind reads from nearby spots. While barcode swapping might also affect an ST experiment during sequencing, spot swapping is a distinct type of contamination.

Below we demonstrate the effect of spot swapping in multiple ST experiments. While it is straightforward to quantify the extent of spot swapping from tissue spots to background spots, assessing the extent of spot swapping within tissue is challenging in most settings without prior information. Toward this end, we consider marker genes where expression is known to be high in particular tissue regions, and low in others. We also conduct a human–mouse chimeric experiment to evaluate the extent of human-specific transcripts in mouse regions, and vice versa.

To adjust for spot swapping in ST experiments, we propose a computational approach called SpotClean, implemented in the R package *R/SpotClean*. Simulations and case study analyses show that SpotClean increases the specificity of marker gene expression, increases the power for identifying differentially expressed

genes, improves the specificity of clusters, and increases the accuracy of spot annotations. The impact of these improvements in studies of breast, pancreatic, and colorectal cancer is also demonstrated.

## Results

**Spot swapping in public datasets**. Figure 1 shows spot swapping from tissue to background in a study of human brain from Maynard et al. [9]. Specifically, Fig. 1b shows that UMI counts at background spots (which are zero in the absence of contamination) are far from zero, with the counts decreasing with increasing distance from the tissue. The distributions of total UMI counts in tissue and background spots show considerable overlap (Fig. 1c); and the expression patterns at tissue spots and nearby background spots are similar, but distinct from distant background spots, as shown for 50 genes in Fig. 1d. As a result of expression similarity between the tissue and nearby background, tissue and background spots often cluster together. This is emphasized in Fig. 1f, where spots on the slide are colored by membership in the graph-based clusters shown in Fig. 1e. As shown, many of the clusters contain spots from the tissue and nearby background. Supplementary Figs. 2–5 show similar results from 16 additional datasets; and Supplementary Table 1 shows that the proportion of UMI counts in background spots ranges from 5% to 20% in most datasets.

Figure 1, Supplementary Figs. 2–5, and Supplementary Table 1 demonstrate that spot swapping occurs from tissue to background. The effect is not explained by inefficient barcode binding, as might be observed in mitochondrial genes or long non-coding RNAs, for example (Supplementary Fig. 6), or by differences in permeabilization times (Supplementary Fig. 7). While spot swapping from tissue to background reduces expression levels at affected spots, a bigger concern is spot swapping from one tissue spot to another, as this confounds downstream analyses.

Evaluating the extent of spot swapping from tissue spot to tissue spot is challenging as it requires information about expected expression of specific genes at specific tissue locations. Toward this end, we first consider tissue-specific marker genes that identify distinct tissue layers in brain[9]. In the absence of spot swapping, expression for a layer-specific marker should be high within that layer, and low (or off) in other layers. When spot swapping occurs, marker expression is relatively high in adjacent layers and decreases with increasing distance from the layer. This is evident with GFAP, for example, a marker known to be up-regulated in white matter (WM) and in the first annotated layer of the dorsolateral prefrontal cortex (Layer1)[9]. Supplementary Figure 8 shows high expression of GFAP in WM and Layer1 spots, as expected, but also relatively high expression in tissue spots adjacent to WM and Layer1, with GFAP expression decreasing as distance from WM (or Layer1) increases. While it is possible that some increase in marker expression in adjacent tissue spots may be due to the presence of WM (or Layer1) cells at those spots, we note that the rate of expression decay into the background spots (where no cells are present) is similar to the rate of decay into adjacent tissue regions. Consequently, the possible presence of WM (or Layer1) cells in adjacent tissue spots is not sufficient to fully explain the observed expression pattern. Similar results are shown for a WM marker, MOBP (Supplementary Fig. 8), as well as additional markers in multiple datasets (Supplementary Fig. 9).

A study of human breast cancer provides another example. Supplementary Fig. 10 shows expression for a highly specific breast cancer marker, ERBB2 (also called HER2). Because of its high specificity (it is typically expressed at a low level in normal breast tissue, but highly expressed in many breast tumors[10]), ERBB2 is used in clinical practice as a target of a number of

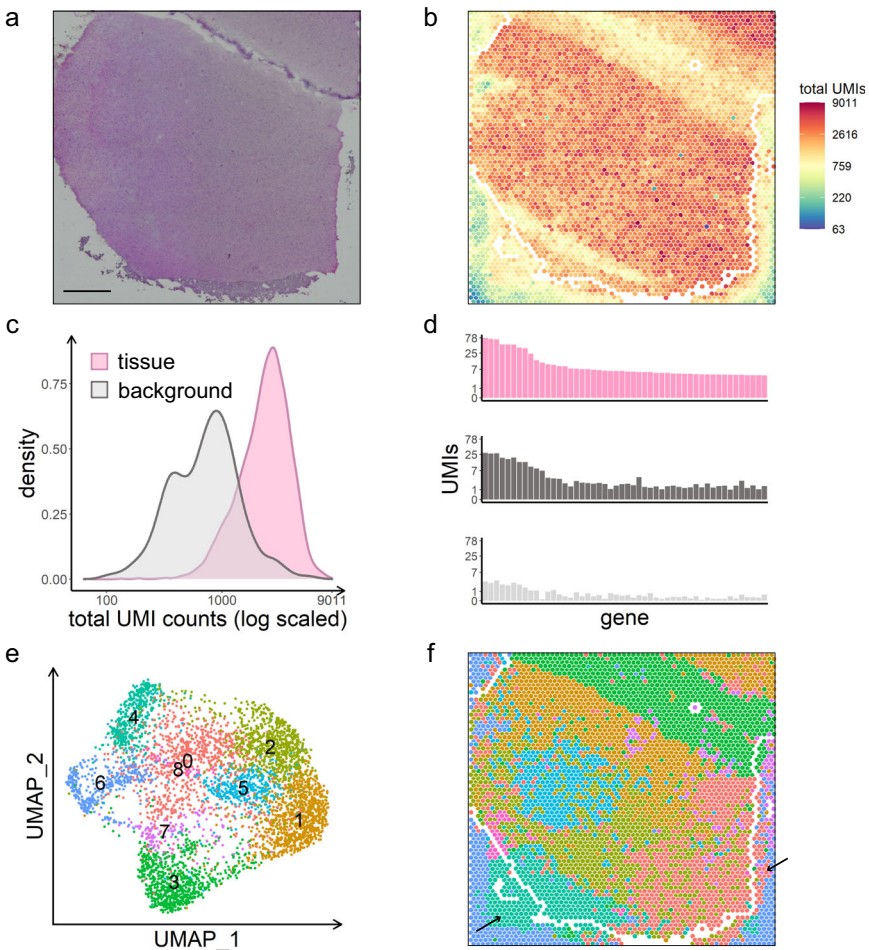

**Fig. 1 Spot swapping in human brain sample LIBD_151507. a** H&E-stained image. Scale bar is 1 mm. **b** Unique molecular identifier (UMI) total counts in the background decrease with increasing distance from the tissue. Tissue and background spot annotations are taken from Maynard et al. [9]. The perimeter delineating tissue and background is shown in white; also shown in white are tissue spots originally called background in Maynard et al. [9] that appear to contain tissue in the H&E image. The spots shown in white have been removed from the summaries shown in **c**–**f** to ensure that the effects shown are not due to spots on the tissue-background boundary, or to tissue in the background. **c** UMI count densities for tissue and background spots show relatively high counts in the background. **d** Counts of the top 50 genes from a select tissue region (upper), from a nearby background region (middle), and from a distant background region (bottom) show the similarity between expression in tissue spots and nearby background spots due to spot swapping from tissue to background, an effect that decreases as distance from the tissue increases. The positions of the three regions are shown in Supplementary Fig. 2. **e** Graph-based clustering of all spots identifies 9 clusters, visualized in uniform manifold approximation and projection (UMAP) plot. **f** Spots on the slide are colored by their cluster membership shown in **e**. Black arrows highlight areas of spot swapping of signal from tissue to background.

therapies[11]. Supplementary Fig. 10 shows high expression of ERBB2 in the tumor tissue, but also high expression in nearby normal tissue that decreases with increasing distance from the tumor. As mentioned above, the increased expression in adjacent normal tissue may be due to the presence of both tumor and normal cells in those spots. However, this is not sufficient to fully explain the effect as the rate of decay from tumor tissue to adjacent normal tissue is similar to the rate of decay from tumor into the background, where no cells are present.

**Experimental validation of spot swapping using chimeric samples**. To more directly quantify the extent of spot swapping, we conducted chimeric experiments where human and mouse tissues were placed contiguously during sample preparation. For each experiment, we annotated the H&E images to identify species-specific regions, and we calculated the proportion of mouse-specific reads in human spots and human-specific reads in mouse spots (Fig. 2, Supplementary Fig. 11). This is a lower bound on the proportion of spot-swapped reads (LPSS) as it does not account for spot swapping within species (e.g. reads from

human spot $t$ bound by probes at human spot $t'$), or for reads in the background. LPSS ranges between 10% and 15% in these experiments (Supplementary Table 1).

Taken together, results from a comparison of tissue and background expression (Fig. 1 and Supplementary Figs. 2–5), analysis of marker genes in brain and breast cancer tissue (Supplementary Figs. 8–10), and the chimeric experiment (Fig. 2, Supplementary Fig. 11, and Supplementary Table 1) demonstrate that spot swapping affects UMI counts in ST experiments. As we show below, this nuisance variability decreases the power and precision of downstream analyses.

**SpotClean is a probabilistic model that adjusts for spot swapping**. To adjust for the effects of spot swapping in ST experiments, we developed SpotClean. SpotClean is based on a probabilistic framework that accommodates spot-swapped reads to provide improved estimates of UMI counts for every gene at each spot. Specifically, SpotClean models gene-specific expression at a given spot as a function of reads present in tissue at that spot, reads that bleed out into other spots, and reads that bleed in from

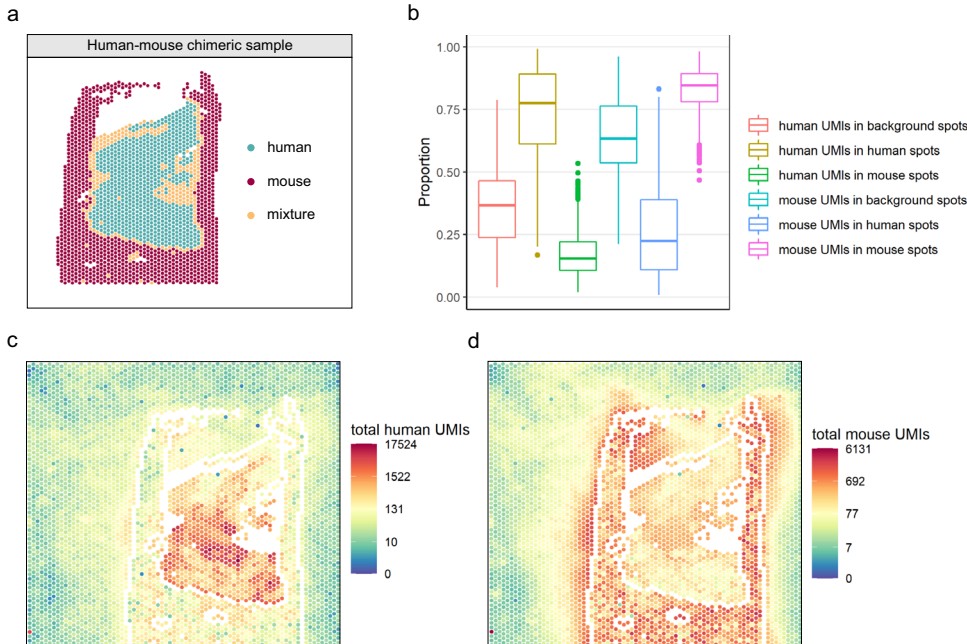

**Fig. 2 Spot swapping in human–mouse chimeric sample HM-1. a** Species annotation of sample HM-1, a chimeric tissue of human skin and mouse duodenum. Spots annotated as mixtures were removed prior to calculating the summaries in **b**–**d** in an effort to ensure that the effects shown are not due to spots containing a mixture of the two species. **b** The spot-specific proportions of spot-swapped UMI counts (human-specific UMIs in background or mouse spots; mouse-specific UMIs in background or human spots). Also shown are the proportion of human-specific UMIs in human spots and mouse-specific UMIs in mouse spots. For the six boxplots from left to right, $n = 2962, 751, 1014, 2962, 751, 1014$ spots, respectively. The lower whisker, lower hinge, line inside box, upper hinge, and upper whisker represent the minimum, lower quartile, median, upper quartile, and maximum calculated without outlier values which are more than 1.5*inter-quartile range away from the hinges and are shown in separate dots. Note that there may be spot-swapped reads in these latter proportions (e.g. reads from human spot $t$ bound by probes at human spot $t'$), but they cannot be identified in this experiment. **c**–**d** The total UMI counts in human-specific genes and mouse-specific genes for HM-1. Similar plots for HM-2 and HM-3 are shown in Supplementary Fig. 11. Tissue spots on the perimeter as well as spots annotated as mixtures were removed prior to calculating the proportions in **b** in an effort to ensure that the effects shown are not due to spots on the tissue-background boundary.

other spots. Bleeding rates and the size of the neighborhood affected are estimated via gradient descent; latent expression levels are estimated using an EM algorithm.

**SpotClean recovers true gene expression, provides more precise estimates of marker gene expression, and improves downstream analyses.** Evaluations were conducted on simulated and case study data. In SimI, contaminated counts are generated assuming that local contamination follows a Gaussian kernel; SimII-IV relax the Gaussian assumption. Supplementary Tables 2–5 show the mean squared error (MSE) between true and decontaminated gene expression in simulated datasets; SpotClean provides better estimates of expression, reducing the MSE by over 20% in most simulations.

The benefits of SpotClean on marker gene estimation and on downstream analyses are also illustrated in case study data. Specifically, Fig. 3a shows that SpotClean improves the specificity of GFAP by maintaining expression levels in WM and Layer1 and reducing spurious expression in the other layers. Supplementary Fig. 9 shows similar results for four additional markers.

We also identified differentially expressed (DE) genes between WM and Layer6 using raw and SpotClean decontaminated data; Fig. 3b and Supplementary Fig. 12 show results for gold-standard genes known to be DE between these layers as discussed in Maynard et al. [9]. In most cases, data processed via SpotClean results in increased fold-changes and smaller $p$-values, further suggesting that SpotClean results in more accurate expression estimates.

Additional results are demonstrated in a study of breast cancer. Figure 4 shows expression for ERBB2 and MUC1, another breast cancer marker, before and after SpotClean. SpotClean increases specificity of these markers by maintaining expression in the tumor regions and reducing expression in the non-tumor regions. It also leads to improved separation of the tumor and non-tumor regions via clustering, as shown visually, and quantified by ARI scores. Similar results are shown in Fig. 5 in a study of pancreatic cancer[12].

**SpotClean reduces the risk of overestimating malignancy in cancer studies.** As the diagnosis and extent and invasiveness of a tumor is typically estimated through evaluation of an H&E image by a pathologist, there is now considerable interest in using ST experiments, which couple the H&E image with molecular profiling data, to improve diagnosis and precision therapy. ST can provide additional information by identifying subtle collections of malignant cells, but accurate spot annotation is required for this information to be useful in clinical practice, and especially so as not to overcall tumor burden. SpotClean demonstrates advantage toward this end. Figure 6a shows spots annotated using SpotClean data versus spots annotated using data that has not been decontaminated via SpotClean for the breast cancer sample discussed above. Compared with the H&E image annotations shown in Fig. 6a, which we consider to be a gold standard, the non-decontaminated data misidentifies many spots as malignant including those containing benign cells surrounding the tumor; the SpotClean decontaminated data more closely resembles identification of malignant cells on the H&E image. Specifically, over 13% of the spots are labeled malignant in the raw, but not

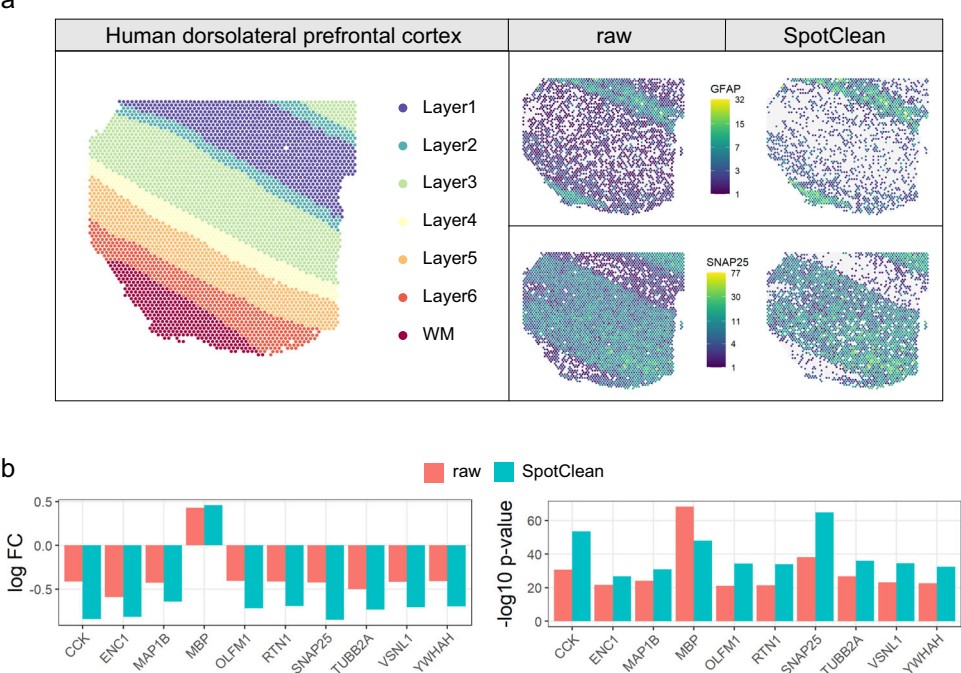

**Fig. 3 SpotClean improves marker specificity in human brain sample LIBD_151507. a** Known annotation of different layers of the human brain sample LIBD_151507 (left); layer-specific marker gene expression in the raw (middle) and SpotClean decontaminated (right) data show that SpotClean provides improved specificity of marker gene expression for GFAP, a marker for WM and Layer1, and for SNAP25, a neuronal marker up-regulated in Layer2–Layer6. **b** An analysis of genes known to be differentially expressed (DE) between WM and Layer6 in raw and SpotClean decontaminated data shows that SpotClean results in increased fold-changes (FC) and smaller *p*-values for the majority of known DE genes.

SpotClean decontaminated, data. Figures 6b and c show that expression in these questionably malignant spots is more similar to spots known to harbor non-malignant cells suggesting that these questionably malignant spots are false calls.

Similar results are shown in Fig. 7 in a study of colorectal cancer where SpotClean decontaminated data leads to improved delineation of tumor and non-tumor regions as evidenced by enhanced tumor malignancy scores in tumor spots, and lower malignancy scores in non-tumor spots, compared with raw data (Fig. 7b). SpotClean also identifies a cluster not identified in previous work (SpotClean tumor cluster 1 shown in Fig. 7c); and multiple analyses suggest that this cluster is a distinct tumor sub-population containing both tumor and tumor-infiltrating immune cells. First, a careful look at the H&E stain shows that this group of spots is non-normal, but distinct from other tumor regions (red boxes, Fig. 7a). Second, 9 of the top 10 genes identified as DE between SpotClean tumor cluster 1 and other tumor clusters are immunoglobulin marker genes (from 74 total DE genes with adjusted *p*-value ≤ 0.01); and immunoglobulin expression for these 9 genes is largely specific to this cluster (Fig. 7d). Finally, the average malignancy score for this group is lower than other tumor clusters, but higher than normal spots, further suggesting that this group of spots contains both tumor cells and tumor-infiltrating immune cells (average malignancy scores at normal, tumor cluster 1, and other tumor spots are 0.384, 0.430, and 0.477, respectively). Taken together, this evidence suggests that cluster 1 identified by SpotClean maintains biologically relevant information and, in this case, provides for a more specific clustering that captures subtle structure present in the tissue.

## Discussion
Common sources of contamination in next-generation sequencing experiments such as barcode swapping[4–6] and ambient RNA

contamination[7,8] have been widely recognized over the past decade (Supplementary Section S1). We here identify spot swapping, a related but distinct form of contamination present in the 10x Visium[1], SpatialTranscriptomics[1], and Slide-seqV2[2] platforms. SpotClean adjusts for the effects of spot swapping using a probabilistic model that accommodates spot-swapped reads to provide improved estimates of gene-specific UMI counts at each spot. SpotClean may be used to obtain improved estimates of expression given data from the 10x Visium or SpatialTranscriptomics platforms; it is not applicable to platforms where background barcodes and/or accurate barcode positions are not provided (e.g. Slide-seqV2).

We have demonstrated the utility of SpotClean to adjust for spot swapping and, in doing so, to provide improved estimates of expression. Since the probability of a spot-specific barcode binding reads from another spot increases as the distance between spots decreases, most of the adjustments made by SpotClean are local (i.e. reads are reassigned from one spot to a nearby spot). Given this, SpotClean will have only a modest impact on some downstream analyses, but a more major impact on others. Specifically, since average expression within a region will remain largely unchanged post SpotClean, downstream analyses that rely on average expression (e.g. DE analyses) will show only slight improvements over the raw data, as shown here. Modest improvements can also be expected for data where clusters are easily separated. However, for more specific analyses and/or more subtle signals, the effects of SpotClean are greater. Specifically, SpotClean provides substantial improvements in marker gene analyses by decreasing expression in regions where markers are known to be lowly expressed, while maintaining expression levels in other regions. In addition, Spot-Clean substantially improves clustering results and spot annotations in situations where regions are not easily separated, which may have important implications for clinical applications of the ST technology (e.g. in cancer diagnosis and staging).

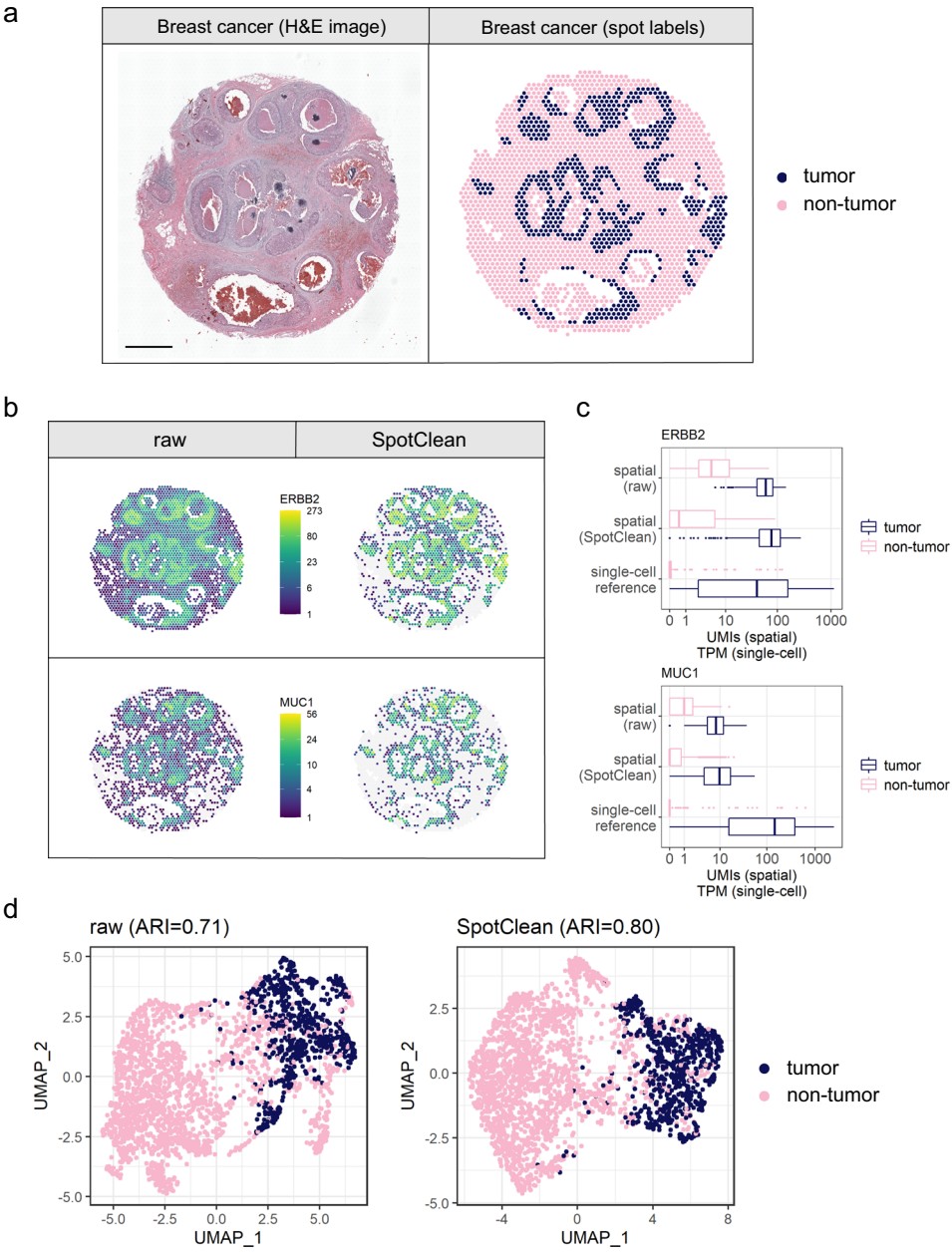

**Fig. 4 SpotClean improves marker specificity in human breast cancer sample human_breast_2. a** H&E image (left) and spots annotated as tumor vs. non-tumor via a pathologist's visual inspection (right). Scale bar is 1 mm. **b** Expression of two tumor-specific markers in the raw (left) and SpotClean decontaminated (right) data. SpotClean increases specificity of these markers by maintaining expression in the tumor regions, and reducing expression in the non-tumor regions. **c** Boxplots of the marker expression shown in **b** as well as the marker expression in a breast cancer single-cell RNA-seq reference dataset[23]. For the six boxplots from top to bottom in both ERBB2 and MUC1, $n = 1843, 675, 1843, 675, 198, 317$ spots or cells. The lower whisker, lower hinge, line inside box, upper hinge, and upper whisker represent the minimum, lower quartile, median, upper quartile, and maximum calculated without outlier values which are more than 1.5*inter-quartile range away from the hinges and are shown in separate dots. **d** UMAP plots generated from raw and SpotClean decontaminated data colored by spot annotations. SpotClean decontaminated data leads to improved separation of the groups, as shown visually, and quantified by the adjusted rand index (ARI) scores which show a 13% improvement in the SpotClean decontaminated data.

In summary, spatial transcriptomics provides unprecedented opportunity to address biological questions and enhance patient care, but artifacts induced by spot swapping must be adjusted for to ensure that maximal information is obtained from these powerful experiments. SpotClean provides for more accurate estimates of expression, thereby improving spot annotations and increasing the power and precision of downstream analyses.

## Methods

**Versions.** The following software and packages were used in the analysis: R-4.0.2; R/SpotClean-0.99.0; R/SoupX-1.5.0; R/celda-1.5.11; R/Seurat-3.2.2; R/scran-1.17.20; R/SPOTlight-0.1.7; R/reticulate-1.16; Python-3.7.4; Python/spatialde-1.1.3; bcl2fastq v2.20.0.422; FastQC-0.11.7; MultiQC-1.9; Space Ranger-1.2.2; Loupe Browser-4.2.0.

**SpotClean.** Let $K$ be the total number of spots, $G$ be the set of genes, $I_t$ be the set of tissue spots with cardinality $|I_t| = K_t$, and $I_b$ be the set of background spots with

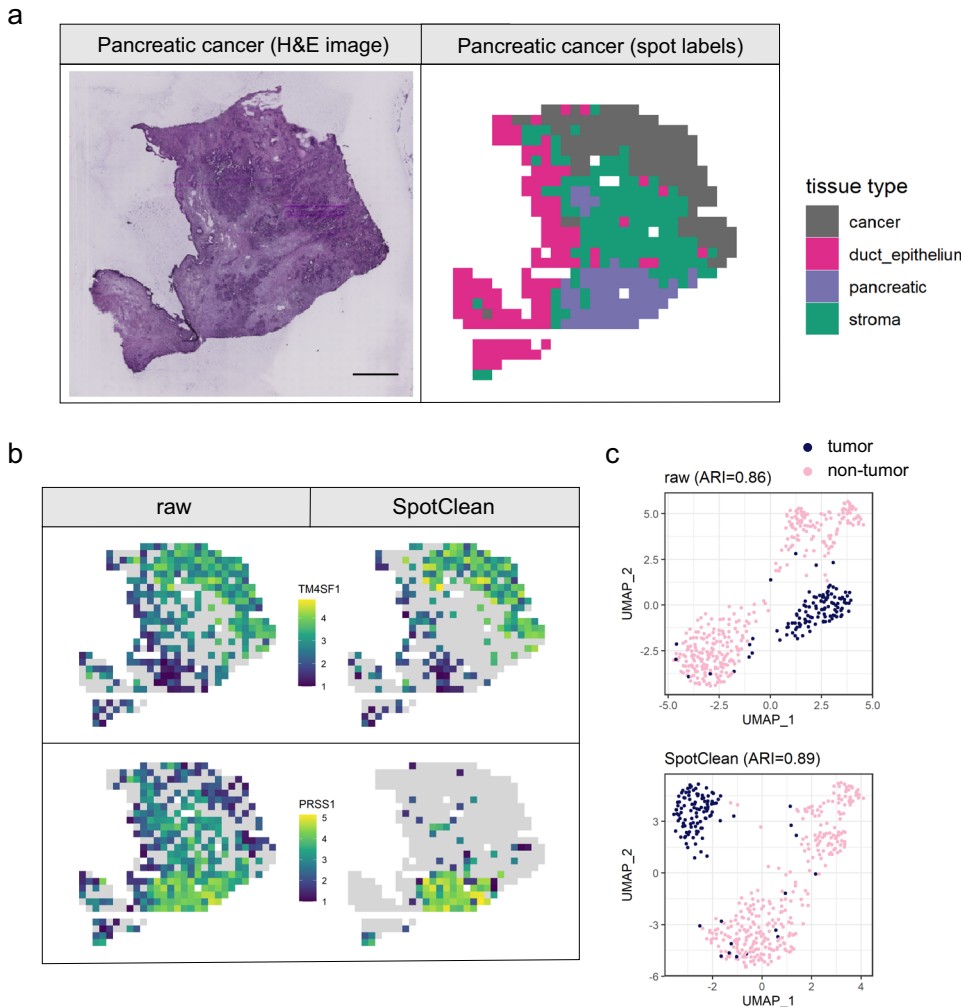

**Fig. 5 SpotClean improves marker specificity in human pancreatic cancer[12] sample PDAC-A. a** H&E image (left) and spots annotated as tumor, duct epithelia, pancreatic tissue, and stroma from the original study (right). Scale bar is 1 mm. **b** The upper panel shows expression of the tumor-specific marker TMSF1 in the raw (left) and SpotClean decontaminated (right) data. SpotClean increases specificity of this marker by maintaining expression in the tumor region, and reducing expression in the non-tumor regions. A pancreatic-specific marker, PRSS1, is shown in the lower panel; as in the upper panel, the specificity of this marker is increased via SpotClean. **c** UMAP plots generated from raw and SpotClean decontaminated data colored by spot annotations (tumor vs. non-tumor). The groups are well separated even in the raw data, but SpotClean decontaminated data leads to slightly improved separation of the groups.

cardinality $|I_b| = K_b$ where $K_t + K_b = K$. $G$ defaults to genes that are highly expressed, highly variable, or both; this default can be relaxed by a user. The true (i.e., uncontaminated) UMI counts are given by $\{Y_{g,t}\}_{g\in G, t\in I_t}$ and observed counts by $\mathcal{D} = \{X_{g,j}\}_{g\in G, j\in I_t\cup I_b}$. As our interest here is to characterize the extent of spot swapping, we introduce the missing variable $B_{g,t,j}$ to be the UMI count for gene $g$ leaving tissue spot $t$ and binding to tissue (or background) spot $j$. Likewise we define $S_{g,t}$ to be the UMI count arising from gene $g$ in tissue spot $t$ that remain at that spot and thus are not subject to bleeding. We decompose $Y_{g,t}$ into a sum: $Y_{g,t} = S_{g,t} + B_{g,t}$, where $B_{g,t} = \sum_{k\in I_t\cup I_b} B_{g,t,k}$ counts all bleed-outs from spot $t$ to other spots $k \neq t$. Extending notation, we set $Y_{g,b} = S_{g,b} = B_{g,b} = 0$ for background spots $b \in I_b$ since background spots do not express RNA. With these missing variables defined, we note that the measured count $X_{g,j} = S_{g,j} + R_{g,j}$ where $R_{g,j} = \sum_{k\in I_t} B_{g,k,j}$ represents UMI counts received at spot $j$ due to spot swapping. We leverage this missing-data formulation by flexibly modeling the component counts with independent Poisson distributions, which are known to be effective for UMI counts[13].

For a collection of spot and gene-specific parameters, as well as global parameters controlling the swapping rates, we parameterize the distributions as: $S_{g,t} \sim \text{Poisson}(\mu_{g,t}(1 - r_\beta))$ and $B_{g,t,j} \sim \text{Poisson}(\mu_{g,t}r_\beta[(1 - r_\gamma)w_{t,j} + r_\gamma\frac{1}{K}])$ where $r_\beta$ is the bleeding rate; $r_\gamma$ is a distal and $1 - r_\gamma$ is a proximal contamination rate. By taking the global bleeding rate $r_\beta \in [0,1]$, it follows that the uncontaminated counts follow: $Y_{g,t} \sim \text{Poisson}(\mu_{g,t})$ for target parameters $\mu_{g,t}$ whose estimates constitute statistical estimates of the uncontaminated counts. Likewise for

measured counts, $X_{g,j} \sim \text{Poisson}(\eta_{g,j})$, for induced gene and spot parameters. We define $w_{t,j}$ by a weighted Gaussian kernel: $w_{t,j} = K(d_{t,j}, \sigma) / \sum_{j'} K(d_{t,j'}, \sigma)$ where $d_{t,j}$ is the physical Euclidean distance between spots $t$ and $j$ measured in pixels in the slide image, $\sigma$ is the kernel bandwidth, and $K(d, \sigma) = e^{(-d^2/2\sigma^2)}$ is a Gaussian kernel[14].

**Parameter estimation.** Plug-in estimates obtained by minimizing the residual sum of squares (RSS) between observed total counts and their expected values are used to estimate $r_\beta$, $r_\gamma$, and $\sigma$. Specifically,

$$(\widehat{r_\beta}, \widehat{r_\gamma}, \hat{\sigma}, \{\widehat{\mu_{\cdot t}}\}_{t\in I_t}) = \operatorname*{argmin}_{r_\beta, r_\gamma, \sigma, \{\mu_{\cdot t}\}_{t\in I_t}} \sum_{j\in I_t\cup I_b} (X_{\cdot j} - \eta_{\cdot j})^2 \qquad (1)$$

where $X_{\cdot j}, \eta_{g,j}, \mu_{\cdot j}$ are the summations of $X_{g,j}, \eta_{g,j}, \mu_{g,j}$ among all genes, respectively. To reduce computational complexity, $\hat{\sigma}$ is taken as the minimum RSS calculated over a grid of candidate values. Explicit gradients are calculated for $r_\beta$ and $r_\gamma$ and estimates are obtained by L-BFGS-B gradient descent[15]. Details are provided in Supplementary Section S2. Since this optimization problem is not necessarily convex, it is important to choose appropriate initial values. For the initial values $\{\mu_{\cdot t}^{(0)}\}_{t\in I_t}$ of $\{\mu_{\cdot t}\}_{t\in I_t}$, we use the observed total UMI counts $\{X_{\cdot t}\}_{t\in I_t}$ in tissue spots and scale them up so that they sum to the total UMIs in the data. The initial bleeding rate, $r_\beta^{(0)}$, is the average expression in background spots divided by the average expression in all spots; and the initial distal contamination rate, $r_\gamma^{(0)}$, is

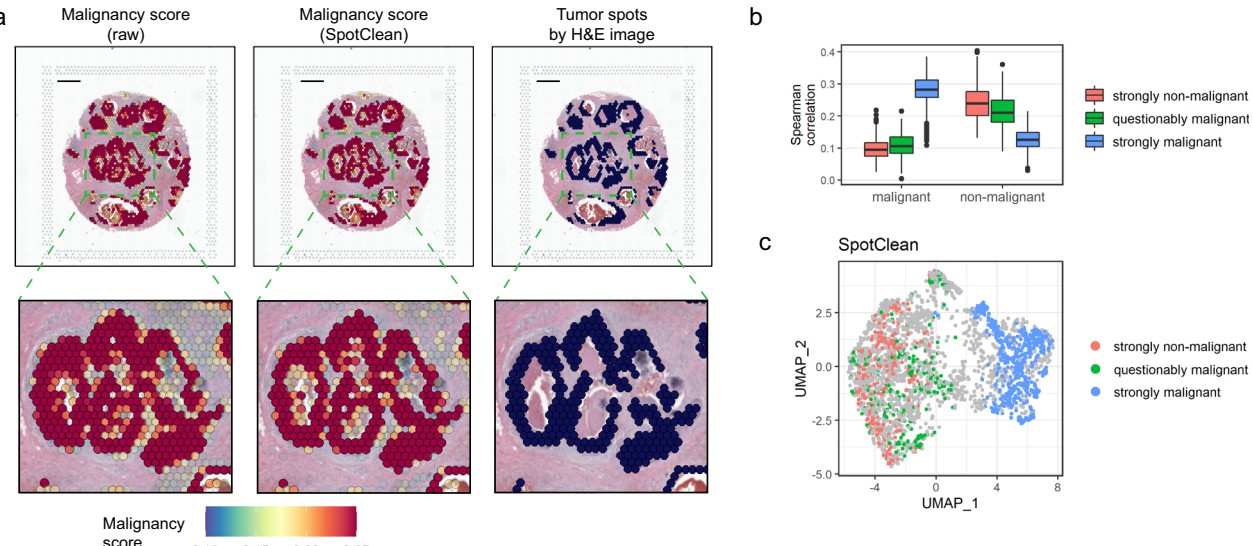

**Fig. 6 SpotClean improves identification of malignant spots in human breast cancer sample human_breast_2. a** Malignant spot composition as estimated via SPOTlight[22] is shown for the raw data (upper left) and SpotClean decontaminated data (upper middle). Scale bar is 1 mm. The raw data identifies many spots as malignant whereas the SpotClean decontaminated data more closely resembles the annotations derived from the H&E image (upper right). The insets highlighted in the upper panel are shown in the lower panel. **b** Spearman correlations between average expression in the malignant scRNA-seq cells and spot-specific expression were calculated. Boxplots of correlations are shown for $n = 265$ strongly non-malignant spots, 216 questionably malignant spots (spots labeled malignant in the raw data, but not the SpotClean decontaminated data), and 546 strongly malignant spots. The lower whisker, lower hinge, line inside box, upper hinge, and upper whisker represent the minimum, lower quartile, median, upper quartile, and maximum calculated without outlier values which are more than 1.5*inter-quartile range away from the hinges and are shown in separate dots. Correlations with non-malignant scRNA-seq cells are also shown. The correlations show that expression in the questionably malignant spots more closely resembles that in non-malignant cells suggesting that the malignant classification in the raw data at these spots is likely false due to spot swapping. **c** The UMAP plot further demonstrates that the questionably malignant spots in the raw data are likely false positives as their expression more closely resembles that at non-malignant spots.

defined by average expression in the 25th–50th percentile of all background spots divided by average expression in all background spots.

With estimates $\hat{r}_\beta, \hat{r}_\gamma, \hat{\sigma}$ of the global parameters, true expression levels $\{\mu_{g,t}\}_{g \in G, t \in I_t}$ are readily estimated using an expectation-maximization (EM) algorithm[16]. Details are provided in Supplementary Section S3. For the initial values of true expressions $\{\mu^{(0)}_{g,t}\}_{g \in G, t \in I_t}$, we use the observed UMI counts $\{X_{g,t}\}_{g \in G, t \in I_t}$ and scale up each gene so that their summations are equal to the gene summations in all spots.

**Estimation of spot-level contamination rate.** For tissue spot $t$, let $c_t$ be the proportion of contaminated UMIs from total observed UMIs. We estimate $c_t$ using the estimated contamination received in $t$ over its estimated contaminated total counts from model fitting:

$$\hat{c}_t = \frac{\hat{E}\left(\sum_{t' \in I_t - \{t\}} \sum_g B_{g,t',t}\right)}{\hat{E}(X_{\cdot t})}.$$

**Minimum number of background spots required for parameter estimation.** Given that the observed data is a single matrix with a fixed number of columns (spots), the number of unknown parameters is proportional to the number of tissue spots. In the extreme case where all spots are covered by tissue, we have more unknown parameters than observed data values. In this case the contaminated expressions are confounded with true expressions, and SpotClean estimation becomes unreliable. We recommend that the input data have at least 25% of spots not occupied by tissue, so that SpotClean has enough information from background spots to reliably estimate contamination.

**Analysis of publicly available case study datasets.** We downloaded UMI count matrices for 16 publicly available datasets, of which 13 came from 10x Visium[1], 1 came from SpatialTranscriptomics[1], and 2 came from Slide-seqV2[2]; links are provided in Supplementary Table 6. For each Visium and SpatialTranscriptomics dataset considered, the count matrix was normalized via scran[17], following the Seurat[18] pipeline for dimension reduction, clustering, and visualization. Seurat functions *FindVariableFeatures(nfeatures = 4000)*, *ScaleData()*, *RunPCA()*, *RunUMAP()*, *FindNeighbors()*, and *FindClusters()* were applied under default settings. For each Slide-seqV2 dataset, we inspected total UMI counts of all spatial barcodes in the raw count matrix.

**Evaluation of spot swapping in mitochondrial and long non-coding RNAs.** To investigate potential differences in bleeding rates in mitochondrial genes and long non-coding RNAs (which may have less efficient barcode binding) vs. other genes, we calculated the proportion of UMI counts in background spots for each gene. Genes having UMI counts greater than 10 were divided into two groups: (1) mitochondrial genes and long non-coding RNAs, (2) the remaining genes. A two-sided Student's $t$-test was conducted between the two groups for each dataset.

**Identification of marker genes and DE genes in the DLPFC data.** Maynard et al. [9] consider spatial expression in the six-layered dorsolateral prefrontal cortex (DLPFC). The authors identified a number of marker genes for distinct layers of the DLPFC. In addition to these, we also considered marker genes from a single-cell RNA-seq study of Alzheimer's disease[19] where markers differentiating between known cell types were identified. The markers shown here were selected from these papers if they were highly expressed (in the upper 25th percentile) in the Maynard et al. [9] datasets. We also evaluate the genes reported as DE between WM and Layer6 in Maynard et al. [9]. We filtered their list of DE genes and considered those genes having FDR $\leq 10^{-4}$. From those, we chose the top 100 highest expressors in the raw data, sorted by fold change, and selected the top 10 for each dataset. For the DE analysis, raw and decontaminated tissue matrices were normalized using scran[17]; for each gene, $p$-values were obtained from a two-sample two-sided $t$-test between the 354 spots in WM and the 486 spots in Layer6. Summary statistics for the tests in Fig. 2b are reported in Supplementary Tables 7 and 8.

**Human–mouse chimeric experiment.** Fresh sections of normal human skin tissue were obtained from two participants (60-year-old and 39-year-old) after obtaining informed consent during routine Dermatologic surgery. Participants were not compensated. Human studies were conducted under a protocol (#2010-0367) approved by the School of Medicine and Public Health Institutional Review Board at the University of Wisconsin. The two participants in this study were randomly recruited in clinic on a single day during routine Dermatologic surgery. No other potential participants were approached because additional tissue was not needed. The participants consented to have their normal residual tissue from Dermatologic surgery used for this study. While this recruitment is biased toward patients who needed a Dermatologic procedure for non-melanoma skin cancer treatment, given that we intentionally sampled normal skin from these patients, we do not anticipate an impact on our results. The covariate-relevant population characteristics are as follows: 60-year-old and 39-year-old patients with current diagnoses of non-melanoma skin cancer requiring Dermatologic surgery for treatment.

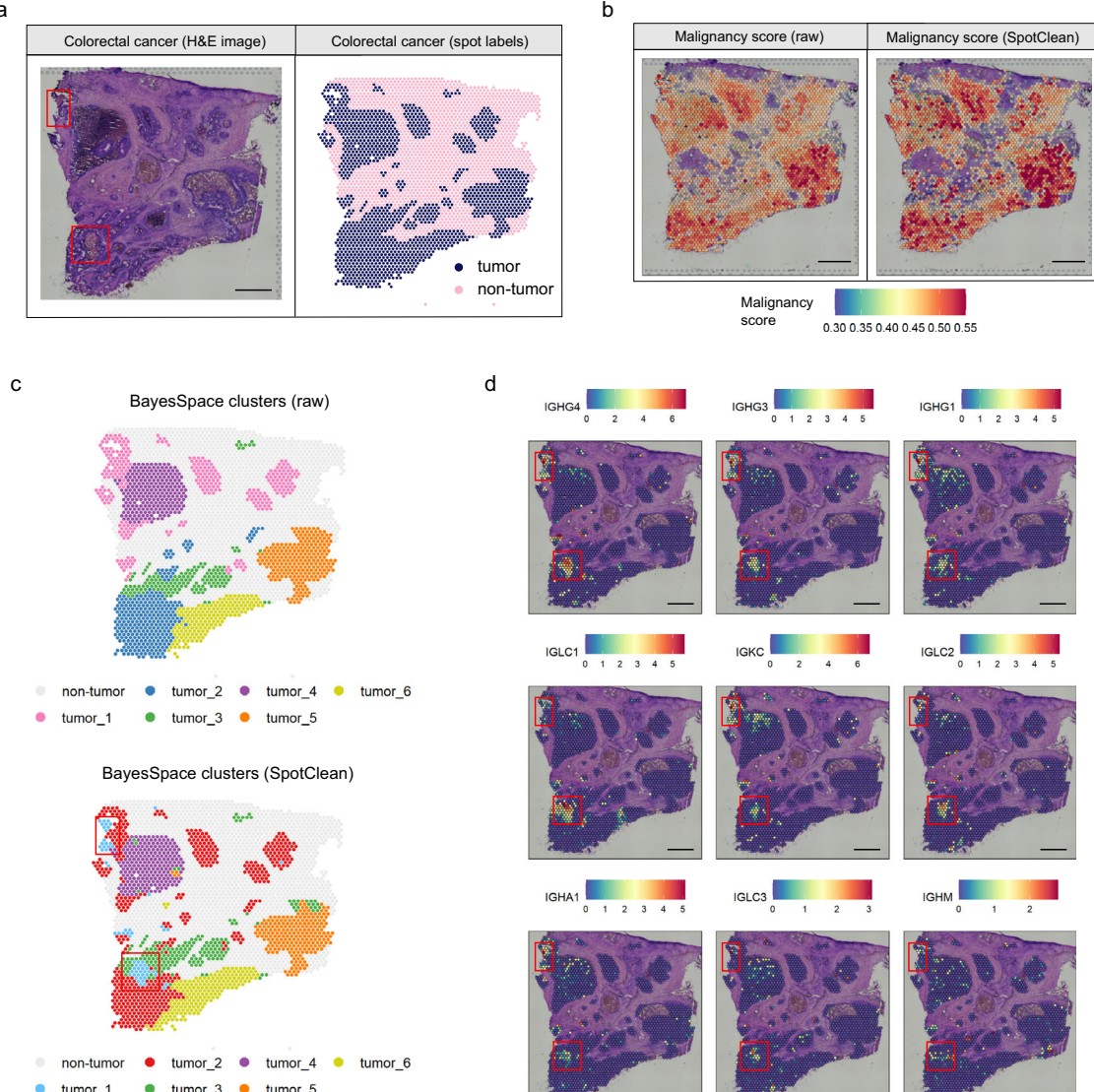

**Fig. 7 SpotClean processed data leads to improved identification of malignant spots and increased specificity in clustering in human colorectal cancer sample human_colorectal. a** H&E image (left) and spots annotated as tumor vs. non-tumor via a pathologist's visual inspection (right). Red boxes highlight the spots belonging to SpotClean's tumor cluster 1 (panel **c**). **b** Malignant spot composition as estimated via SPOTlight[22] is shown for the raw (left) and SpotClean decontaminated data (right). SpotClean results in higher malignancy scores in tumor regions, and lower in normal regions. **c** BayesSpace[25] clustering for the raw data (top) and SpotClean decontaminated data (bottom). The SpotClean decontaminated data identifies a cluster not identified in previous work (SpotClean tumor_1, red boxes). The SpotClean tumor_1 spots are distinct on the H&E image (red boxes in **a**) and likely contain tumor-infiltrating immune cells as evidenced by high expression in the immunoglobulin markers shown in **d**. Scale bars in **a**, **b**, **d** are 1 mm.

Fresh mouse tissue was harvested on the same day. Animal studies were conducted under a protocol (#M5131) approved by the School of Medicine and Public Health Institutional Animal Care and Use Committee at the University of Wisconsin in compliance with policies established by the Office of Laboratory Animal Welfare at the National Institutes of Health. All animals were housed in a specific pathogen-free facility and fed a standard chow diet containing 4% fat. The set point for temperature was 72 °F and the set point for humidity was 30%. The dark/light cycle was 12:12.

Three chimeric tissue blocks were then prepared under cold conditions as follows and frozen over a bed of dry ice and stored at −80 °C in optimal tissue cutting (OCT) medium until they were ready to use:

HM-1: Duodenum from a 10-week-old C57BL/6J male mouse as casing to a 4 mm punch section of human skin from a 60-year-old participant.

HM-2: Colon from a 10-week-old C57BL/6J male mouse as casing to a 4 mm punch section of human skin from a 39-year-old participant.

HM-3: Heart from a 10-week-old C57BL/6J male mouse encasing a 4 mm punch section of human skin from a 39-year-old participant.

The Visium Spatial Tissue Optimization Slide & Reagent kit (10X Genomics Cat no. 1000193) was used to optimize permeabilization conditions for the chimeric tissue according to manufacturer's protocol and yielded an optimal tissue permeabilization time of 12 min. The Visium Spatial Gene Expression Slide & Reagent kit (10X Genomics Cat no. 1000184) was used to generate sequencing libraries. Sections were cut at 10 μm thickness and mounted onto Visium slide capture areas, stained with H&E, digitally imaged with an Aperio AT2 scanner, and then permeabilized for library preparation. Sequencing libraries were prepared following the manufacturer's protocol. Initial quality control of the libraries was by analysis of 2 × 150 MiSeq data for each sample. The libraries were then sequenced on a NovaSeq 6000 (Illumina), with 29 bases from read 1 and 101 from read 2, at a depth of 500k–600k reads per spot. The actual depth was 455,652, 440,024, 538,709 reads per spot for sample HM-1, HM-2, HM-3, respectively.

**Alignment and pre-processing in the chimeric experiment.** Raw FASTQ files were generated using bcl2fastq (Illumina, Inc.). The sequencing quality of each sample was evaluated using FastQC[20] and MultiQC[21]. All FASTQ files passed quality control. Tissue images were manually aligned using the Loupe Browser. Reads were aligned to the GRCh38+mm10 reference genome (refdata-gex-GRCh38-and-mm10-2020-A from https://support.10xgenomics.com/single-cell-gene-expression/software/downloads/latest) and gene expression was quantified

using Space Ranger under default parameters. Following alignment, we considered only those reads labeled confidently mapped by SpaceRanger; confidently mapped reads are reads that map uniquely to a gene. We refer to a gene as a human gene if it has prefix GRCh38; a mouse gene has prefix mm10. UMI counts were normalized for differences in total counts across species by scaling total UMI counts in mouse to match total UMI counts in human. Genes having average expression <0.01 were removed.

**Human and mouse tissue spot annotation in the chimeric experiment**. Tissue spots were labeled as human, mouse, or histopathological mixture based on visual inspection of the H&E images. A histopathological mixture spot is one with tissue contributions from both species that can be visually verified in the H&E-stained image. A pure human or pure mouse spot was relabeled as a computational mixture spot if the spot label differed from the majority of UMIs. Specifically, a human (or mouse) spot was labeled as a computational mixture if the total UMI counts from mouse (human) exceeded the median of total UMI counts across all mouse spots (human spots). Background spots are defined as those spots on the slide outside the tissue region (not annotated as human, mouse, or mixture). Both histopathological or computational mixture spots were removed prior to analyses in an effort to ensure that the effects shown are not due to spots containing a mixture of the two species.

**Lower bound on the proportion of spot-swapped reads (LPSS)**. Spot-swapped reads include reads from one tissue spot binding background probes (tissue-to-background) as well as reads at one tissue spot binding probes at another tissue spot (tissue-to-tissue). It is not possible to directly measure tissue-to-tissue swapping in most cases. However, the chimeric experiment provides some insight into the extent of spot swapping tissue-to-tissue. We define LPSS in the chimeric experiment as the proportion of misclassified reads (mouse reads in human spots and human reads in mouse spots). This is a lower bound as it does not account for spot swapping within species (e.g. reads from human spot $t$ bound by probes at human spot $t'$) or for reads in the background.

**Cell type decomposition using SPOTlight**. For cell type decomposition, we applied SPOTlight[22] to the Visium human breast cancer data and the Visium human colorectal cancer data (referred to here as human_breast_2 and human_colorectal; details on these data are provided in Supplementary Table 6). SPOTlight[22] requires single-cell RNA-seq data to use as a reference; for this, we used the human breast cancer single-cell RNA-seq data from Chung et al. [23] and the human colorectal cancer single-cell RNA-seq data from Li et al. [24]. SPOTlight[22] was applied to the raw data under default settings to estimate the cell type composition of every spot; SPOTlight[22] was also applied to the SpotClean decontaminated data under default settings. Note that since tumor cell populations are heterogeneous, and spots contain multiple cells, most spots containing malignant cells will also contain non-malignant cells. A spot's malignancy score is defined to be the proportion of malignant cells estimated by SPOTlight[22].

Following clinical practice, we label a spot as malignant if there is any evidence of malignancy. Specifically, we annotate spots as malignant if the estimated malignant cell composition exceeds 10%, which corresponds to ~1 malignant cell in the spot since the estimated number of cells in a spot is ~10 in Visium data[22]. We further define non-malignant spots as strongly non-malignant if the non-malignant cell composition exceeds 95%, and strongly malignant if the malignant cell composition exceeds 30% in both raw and decontaminated data. Questionably malignant is used to refer to spots called malignant in the raw data, but not the SpotClean decontaminated data.

**Identification of DE genes in the colorectal data**. DE analysis was performed using Seurat's pipeline with the Wilcoxon Rank sum test and standard defaults; $p$-values are adjusted using the Bonferroni correction.

**Correlations with single-cell data**. For the breast cancer data, Spearman correlations between the expression of each spot and the average expression of malignant cells in the reference single-cell data were calculated to measure the similarity of each spot group (strongly non-malignant, strongly malignant, or questionably malignant) to malignant cells; the same was done to measure similarity of each spot group to non-malignant cells. Boxplots in Fig. 4c demonstrate the median, upper and lower quartile, range without outliers, and outlier values of the Spearman correlations for each group of spots using default plotting functions.

**Clustering and ARI**. For each cancer case study analysis (breast, pancreatic, and colorectal), the Seurat pipeline was applied under default settings to the raw and decontaminated data to produce UMAP plots. For the breast cancer data and the pancreatic cancer data, tumor spots were clustered using $k$-means clustering ($k = 2$) of the top 50 PCs calculated in the Seurat pipeline. For the colorectal cancer data, tumor spots were clustered using BayesSpace[25] under default settings. In the H&E image, tissue spots were labeled as tumor and non-tumor based on visual inspection. The adjusted rand indexes (ARI) were calculated between cluster labels and tumor/non-tumor labels.

**Simulations**. SimI simulates the spot swapping effect to get contaminated UMI counts given an input dataset. Specifically, starting from an input UMI count matrix of real data, 3000 genes with highest total UMI counts were selected. Expression for these genes was scaled to target the same average UMI total counts (average taken over spots) across input datasets. Denote the resulting matrix by $\{\mu_{g,t}\}_{t \in I_t}$. The bleeding rate $r_\beta$ and distal contamination rate $r_\gamma$ were estimated from the input data, using the same approach as described for obtaining initial values in SpotClean. The spot distances $\{d_{t,j}\}_{t \in I_t, j \in I_t \cup I_b}$ were calculated based on the spot coordinates in the H&E image of the input dataset; the contamination radius, $\sigma$, was set to 10; and the weights which describe the proportion of UMIs swapping locally from tissue spot $t$ to any spot $j$, $w_{t,j}$, is given by a Gaussian kernel. The expected contamination of gene $g$ from tissue spot $t$ to spot $j$ is then given by $\mu_{g,t} r_\beta [(1 - r_\gamma) w_{t,j} + r_\gamma \frac{1}{K}]$. Summing contamination from all tissue spots to spot $j$ and adding the UMIs that stay at $j$, $\mu_{g,j}(1 - r_\beta)$, gives the expected observed expression $\eta_{g,j}$. Simulated counts for gene $g$ in spot $j$ are sampled from Poisson($\eta_{g,j}$).

Additional simulations are similar, but for SimII, SimIII, and SimIV the proximal contamination weights are given by a Linear, Laplace, and Cauchy kernel, respectively. This allows us to investigate the extent to which SpotClean is robust to departures from the Gaussian kernel assumption.

For SimV, starting from a UMI count matrix of real data, we select the top 5000 most highly expressed genes; any gene having average expression <0.1 is removed. SpatialDE[26] is then applied using default settings; the top 500 highest expressed genes with $q$-value ≤ 0.01 are identified as true spatially variable (SV) genes. For each SV gene, we simulate a matched non-SV gene by sampling independent Poisson counts parameterized by the average expression of the SV gene.

**Application of SoupX, DecontX, and SpotClean**. Default parameters were used for SpotClean and DecontX. Since SoupX requires manual input of clusters, we first applied the Seurat pipeline on the raw tissue UMI count matrix to get cluster labels, with functions *NormalizeData()*, *FindVariableFeatures()*, *ScaleData()*, *RunPCA()*, *FindNeighbors()*, *FindClusters()* applied under default settings. Parameters for SoupX (*soupRange* in *estimateSoup()*, *tfidfMin*, and *soupQuantile* in *autoEstCont()*) were manually tuned when the default settings failed. Some datasets did not run even after parameter tuning; results from these datasets are marked as NA. SpotClean decontaminates genes with average expression above 1, high variance as determined by Seurat's *FindVariableFeatures()* function, or both. All methods were applied to these same set of genes. In the simulated data, we force all methods to decontaminate all genes since there are relatively few (1000 or 3000 genes depending on the simulation).

**Reporting summary**. Further information on research design is available in the Nature Research Reporting Summary linked to this article.

## Data availability

Raw sequence data for the three human–mouse chimeric experiments are available at GEO (accession number: GSE178221). Links to 16 public spatial transcriptomics datasets are available in Supplementary Table 6. The human breast cancer single-cell RNA-seq data from Chung et al. [23] is available at GEO (accession number: GSE75688). The human colorectal cancer single-cell RNA-seq data from Li et al. [24] is available at GEO (accession number: GSE81861). Additional datasets used to investigate permeabilization times are available at GEO (accession numbers: GSE169749, GSE178361, GSE188888, GSE190595, and GSE193460). Processed data for reproducing results in our studies are available at Zenodo[27]. The GRCh38+mm10 reference genome is available at 10x Genomics (refdata-gex-GRCh38-and-mm10-2020-A).

## Code availability

The R package *SpotClean* is available at https://github.com/zijianni/SpotClean [28] and will be submitted to Bioconductor. Codes for simulation and case study data analyses can be found at https://github.com/zijianni/codes_for_SpotClean_paper [29].

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

## Acknowledgements

This work was supported by the following fundings: NIH GM102756 (Z.N., C.K.), NIH UL1TR002373 (A.P., B.A.D.), 2020 UW-ICTR Translational Pilot Award (A.P., L.M.A., B.A.D.), NIH/NCI 1 R01 CA220004-01 (R.B.H.), 2020 Dermatology Foundation Pediatric Dermatology Career Development Award (L.M.A.), 2019 Sturge Weber Foundation Lisa's Research Award (L.M.A.), NSF 2023239-DMS (M.A.N.), NIH 1P01CA250972-01 (M.A.N.), NIH 1P50HD105353-01 (M.A.N.). The authors thank the University of Wisconsin Translational Research Initiatives in Pathology (TRIP) laboratory for assistance with sample preparation (P30 CA014520 and S10 OD023526), the University of Wisconsin Biotechnology Center DNA Sequencing Facility for providing RNA sequencing facilities and services, and the University of Wisconsin Biomedical Model Research Services (BRMS) for providing housing and care for the research animals used in this study.

## Author contributions

Z.N. discovered the spot swapping artifact. Z.N. and C.K. designed the research and wrote the first version of the manuscript. Z.N., C.K., and M.A.N. developed the SpotClean method. A.P. and R.B.H. designed the chimeric samples and conducted the chimeric experiments. Z.N. and S.C. conducted simulations and quality control evaluations. Z.N., S.C. and C.K. built and tested the R package. All authors contributed to writing the manuscript.

## Competing interests

The authors declare no competing interests.
