## [Peer Review File · Nature Communications]

Reviewers' Comments:

Reviewer #1:

Remarks to the Author:

In this manuscript, Ni and colleagues aim to improve the sensitivity of analyses relating to Spatial Transcriptomics (ST). The authors propose a novel computational method to adjust for the transfer of transcripts from specific regions in the tissue to barcoded spots adjacent to those regions. Specifically, the authors present a computational method to correct for diffusing transcripts (UMIs) between capture regions in spatial transcriptomics data, which the authors refer to as spot swapping. The method is based on a probabilistic model and aims to increase the precision or sensitivity for downstream computational analyses. The authors present a convincing computational approach which they validate on a multiplicity of biologically and clinically relevant tissue samples. ST is a new method and although it is currently not commonly implemented in clinical pathology (e.g. cancer diagnostics), it is not unlikely that it could be in the future, for example linked to personalized medicine.

Thus, the method described within this manuscript provides an important tool for ST analyses and would be a valuable contribution to the field. However, there are a few open questions and comments outlined below that the authors should address in a revision.

1) The authors' claim, with regards to e.g. the reduced overestimations of malignancy in cancer tissue sections, appear somewhat over-confident in relation to the experimental data present. This was observed in the text throughout the manuscript. This reviewer therefore suggests to either strengthen the claims by further experimental evidence or to rephrase the respective parts in the manuscript accordingly (e.g. lines 183 - 188 and 190 etc.).

2) In Figure 1 the authors show the UMI count densities for tissue and background in panel c) as well as in panel d). Assuming the plots are generated using raw count data, was the data filtered for Inc- and mitochondrial transcripts beforehand? These often appear in the background and the tissue borders in ST experiments, possibly due to the less efficient binding to Poly-A capture regions and are often considered artifacts. If the authors observe higher bleeding rates from Inc- and mitochondrial genes from the tissue in the background it might be advisable to account for this in the model, e.g. by including a weighted parameter for the gene biotype.

3) In lines 92-93 the authors write: "Figure 1, Supplementary Figures 2-5, and Supplementary Table 1 demonstrate that spot swapping occurs from tissue to background. While this reduces tissue signal, thereby reducing the power of the experiment". However, the visualization of UMI counts on the tissue coordinates does not imply that the occurrence of UMIs in the background actually reduces the signal in the tissue, thus it would be good if the authors can support this claim in greater detail.

4) In Figure 2 the authors evaluate spot-swapping events in chimeric tissue. This represents an interesting approach to test for spot-swapping/ mRNA diffusion in ST experiments. However, the human tissue analyzed by the authors seems to be surrounded entirely by mouse tissue, less a small part at the top of the human tissue. Therefore, it would be helpful to know how background spots, especially with regards to the human tissue in the chimeric experiment, were selected given that most of the background spots (spots adjacent to the human tissue) actually represent mouse tissue. Background spot selection does not become entirely clear as spot-swapping events are reported in proportions (Figure 2b).

5) Between lines 151 - 155 the authors describe the performance evaluations on simulated and case data.

a) While they describe five different simulations in the methods section (from line 519), they only display the results of four simulations in the referenced supplementary tables, with SimV missing in the text and tables.

b) In addition, it is confusing why the results of each simulation differ, if the same dataset was taken to perform the simulations for decontamination. If a subset of expression values is used as input by each simulation it would be more reproducible to use the same seed. This way the results

of the comparison of the different SpotClean simulations and the existing decontamination methods (SoupX and DecontX) could be presented altogether.

c) Further, it is not clear in the manuscript if the authors compared their method with other existing methods. The statement "Supplementary Tables 2-5 show the mean squared error (MSE) between true and 154 decontaminated gene expression in simulated datasets; SpotClean provides better estimates of 155 expression, reducing the MSE by over 20% in most simulations." gives the impression that comparisons were only performed between different SpotClean simulations and does not specify in comparison to which decontamination methods/simulations the MSE is reduced by over 20%. In my opinion - it would be important to elaborate and clarify this statement as the authors are directly comparing the performance of their model with published and peer reviewed methods.

6) The authors perform simulations in order to test whether their model outperforms existing methods for removing contaminations of ambient RNA in single cell datasets (SoupX and DecontX), in the spatial context this may underlie different distributions.

a) However, from the manuscript it is understood that the authors assume a Poisson distribution of uncontaminated counts, thus it would be good to elaborate why multiple simulations were performed.

b) Further, the authors compared the decontamination methods on spatial data. It would be interesting to see how SpotClean performs on scSeq data, for which the other two algorithms (SoupX and DecontX) were designed, in order to potentially highlight the strength of SpotClean in the spatial context in comparison to other decontamination methods. To this end the authors should attempt to run SpotClean on publicly available single cell data and compare the performance to the referenced SoupX and DecontX.

7) If understood correctly, SpotClean-decontamination is performed before clustering/factorization and differential gene expression analysis (DGEA) for clusters. The authors report that marker genes within breast cancer samples show higher fold changes and significance (p-values) in the respective clusters (line 167 -172).

a) Do these observations also have an impact on the general number of genes/ gene composition of the individual clusters and if so how is the general clustering affected? Are for example more or less clusters identified after decontamination, inevitably changing the composition of some clusters and how does this impact the biological interpretation of the data?

b) The authors mention generating ARI scores to compare the clustering results but it does not become entirely clear what the input for the ARI is. I am assuming the authors are comparing malignant spot annotations from visual annotations (of the H&E image) and the annotations of malignant cancer spots based on transcription. It would be essential to elaborate what exactly was compared for the ARIs.

c) Further the ARIs presented in Figure 7 d are both very low, indicating close to random assignment of spots to clusters (In this case malignant and non-malignant). Further the difference between the ARIs only has a magnitude of 0.05) which does not appear very convincing in that the decontaminated data is assigning spots notably better to the expected malignant cluster. This observation also applies to Figure 5, where the general ARIs are more convincing (0.86 and 0.89) but the difference between the ARIs is even lower (0.03). In the case of Figure 4 ARI values are most convincing but it might be advisable to consider an alternative/additional measure to compare clustering/correct spot assignment performance.

d) More generally, it is questionable if using ARI to enable quantifying the improved separation of the tumor and non-tumor regions, as stated between line 169 and line 171 as well as line 272 - 273 is an accurate strategy. An ARI is a comparative measure to evaluate if two cluster results are similar to each other. Based on the above described concern about the low magnitudes between ARI values, the authors should justify the use of ARIs to quantify the improved clustering results further.

8) In Figure 7 the authors show the effect of decontamination on clustering results of raw Visium spatial data. Panel c shows the assignment of 6 clusters, including 5 tumor clusters for the raw data but only 4 tumor clusters for the decontaminated data. How do these clusters differ in their total gene expression profiles? Differences between tumors within the same tissue are another essential quality (apart from the position of the tumor) to ensure the most optimal treatment.

a) Based on their claim that "SpotClean decontaminated data leads to improved delineation of

tumor and non-tumor regions and improved identification of clusters.”, how are the authors able to support their claim that the clustering performed after decontamination using SPOTClean resembles the actual tumor biology better than the raw data? The evidence needs to be explained more explicitly.

b) Further validation of differences between the tumor clusters would be necessary, this could for example be done by the analysis of markers for certain tumor subtypes or a more detailed description of the biological differences between the identified tumor clusters.

9) The authors convincingly demonstrate that the method they developed is able to reduce noise signals in spatial data. This is mainly presented by spatial expression profiles of known spatial marker genes (Figure 3,4 and additional Supplementary Figures). However, as the method is removing noise and increased fold-changes and smaller p-values for DEGs, it would be interesting to evaluate whether the method finds additional DEGs between spatial clusters, or if some genes that are considered differentially expressed before decontamination disappear after decontamination.

10) From the experimental perspective, permeabilization efficiency plays a crucial role for the success of a Visium or Spatial Transcriptomics experiment. This permeabilization needs to be optimized for different tissue types and over-permeabilization can lead to higher diffusion rates between spots, i.e. spot-swapping. Have the authors observed substantial differences of bleeding rates r_β and contamination rates r_γ between individual tissue sections and/or tissue types. This could, in addition to the suggested improvements of the clustering and DGEA results, serve as a valuable quality control measurement for the resulting sequencing data and increase the relevance of the suggested method.

11) From line 351 - line 375, the authors describe the SpotClean method in detail. They describe variable G to be a set of genes. It would be helpful to elaborate how this set of genes G is determined, i.e. how are the genes selected, how large is the set of genes, etc. - are these based on e.g. a specific number of variable genes in the count data?

Reviewer #2:

Remarks to the Author:

This is a very interesting work that delivers an important message for the spatial transcriptome field: the signal you get from one spot is a weighted average of nearby spots. The authors have used multiple publicly available datasets and their own data (by combining human and mouse samples into one slide) to demonstrate this. They have also developed a computational method to correct such signal mixing and demonstrate its effectiveness. Overall, this is an important work that will have impact. In some situations, such as downstream analyses relying on average expression (as pointed out by the authors), such correction of spot mixture may not be crucial, but it is still important to know this when interpreting the results.

I have some minor comments on the method details.

1. For the lower bound on the proportion of spot-swapped reads (LPSS), the estimate from the chimeric experiments was 10-15%. Does it mean that 10-15% of UMI from a spot is from nearby spots? It would be helpful to clarify how is this being calculated. Since this is a lower bound and it may not be unreasonable to believe that twice amount of swap happen (e.g., human to human or mouse to mouse), which gives 20-30% of UMI per spot are from nearby spots. Is that right? Is this estimate consistent with other data?

2. what is the identifiability requirement of the model? or in other words, what are required to obtain reliable estimates. For example, if all the spots are homogeneous (but there are still backgrounds), could the spotClean method work?

3. at line 368, " r_γ is a distal and $1 - r_\gamma$ is a proximal contamination rate”, the distal means a constant bleeding to all the K spots? It is hard to imagine that one spot can bleed somewhere faraway?

4. parameters are estimated by minimizing residual sum squares. It is reasonable, but somewhat unusual for a Poisson model since variance increases with mean. Is it more appropriate to use Poisson likelihood as objective function?

We thank the reviewers for their careful review; their detailed comments have helped to improve the manuscript. Below we provide a point-by-point response, with reviewer comments given in bold.

Reviewer #1

In this manuscript, Ni and colleagues aim to improve the sensitivity of analyses relating to Spatial Transcriptomics (ST). The authors propose a novel computational method to adjust for the transfer of transcripts from specific regions in the tissue to barcoded spots adjacent to those regions.

Specifically, the authors present a computational method to correct for diffusing transcripts (UMIs) between capture regions in spatial transcriptomics data, which the authors refer to as spot swapping. The method is based on a probabilistic model and aims to increase the precision or sensitivity for downstream computational analyses. The authors present a convincing computational approach which they validate on a multiplicity of biologically and clinically relevant tissue samples. ST is a new method and although it is currently not commonly implemented in clinical pathology (e.g. cancer diagnostics), it is not unlikely that it could be in the future, for example linked to personalized medicine.

Thus, the method described within this manuscript provides an important tool for ST analyses and would be a valuable contribution to the field. However, there are a few open questions and comments outlined below that the authors should address in a revision.

Thanks!

1) The authors' claim, with regards to e.g. the reduced overestimations of malignancy in cancer tissue sections, appear somewhat over-confident in relation to the experimental data present. This was observed in the text throughout the manuscript. This reviewer therefore suggests to either strengthen the claims by further experimental evidence or to rephrase the respective parts in the manuscript accordingly (e.g. lines 183 - 188 and 190 etc.).

We regret that the results sound over-confident. Perhaps we should have made it more clear that we consider the H&E image annotation to be the gold standard, and so claims such as "The non-decontaminated data misidentifies many spots as malignant" and "the spots labeled malignant in the raw data are likely false calls" are relative to the H&E image. Specifically, if a spot is identified as malignant using the raw data, but it's not annotated as malignant in the H&E image, we consider that a false call. Defined in this way, we found 13% of the spots to be false calls. Comparison to the H&E image also underlies the other main statement in that paragraph which is "SpotClean reduces noise by correctly assigning spot-swapped reads from malignant (and non-malignant) cells back to malignant (non-malignant) spots." We have rephrased the section to read as follows:

“Figure 6a shows spots annotated using SpotClean data versus spots annotated using data that has not been decontaminated via SpotClean for the breast cancer sample discussed above. Compared with the H&E image annotations shown in Figure 6a, which we consider to be a gold standard, the non-decontaminated data misidentifies many spots as malignant including those containing benign cells surrounding the tumor; the SpotClean decontaminated data more closely resembles identification of malignant cells on the H&E image. Specifically, over 13% of the spots are labeled malignant in the raw, but not SpotClean decontaminated, data. Figures 6b and 6c show that expression in these "questionably malignant" spots is more similar to spots known to harbor non-malignant cells suggesting that these questionably malignant spots are false calls.”

2) In Figure 1 the authors show the UMI count densities for tissue and background in panel c) as well as in panel d). Assuming the plots are generated using raw count data, was the data filtered for Inc- and mitochondrial transcripts beforehand? These often appear in the background and the tissue borders in ST experiments, possibly due to the less efficient binding to Poly-A capture regions and are often considered artifacts. If the authors observe higher bleeding rates from Inc- and mitochondrial genes from the tissue in the background it might be advisable to account for this in the model, e.g. by including a weighted parameter for the gene biotype.

Thank you! We had not appreciated this and, consequently, the data was not filtered for Inc- and mitochondrial transcripts beforehand. We definitely agree that if there are higher bleeding rates for these sets of transcripts, it would be helpful to account for it in SpotClean. Having now taken a look, it seems that there is no clear pattern in the datasets that we've considered.

Specifically, below we show boxplots of the estimated bleeding rates for genes having at least 10 counts in the background (we did not consider genes with fewer than 10 counts since the variability of estimated bleeding rates is too high, and the estimation is less reliable) in the 12 publicly available datasets and the chimeric experiments that we used in the manuscript. A gene-specific bleeding rate is estimated using the proportion of counts in background spots (a bleeding rate is zero for a given gene if no counts are in the background and 1 if all counts are in the background).

Each panel has two boxplots; the left (salmon) corresponds to the mitochondrial and long noncoding RNAs (mt_Inc); the right (turquoise) corresponds to the remaining genes.

Figure 1: Boxplots of estimated bleeding rates for mitochondrial and long noncoding RNAs (mt_Inc) and the remaining genes (not_mt_Inc) in 12 publicly available datasets as well as the chimeric experiments.

Results show that there is no consistent trend. We do observe higher bleeding rates in mitochondrial and long noncoding RNAs for three of the datasets (human lymph node; breast cancer; and mouse brain). However, the other datasets show comparable bleeding rates, or bleeding rates that are higher in the non_mt_lnc genes.

3) In lines 92-93 the authors write: “Figure 1, Supplementary Figures 2-5, and Supplementary Table 1 demonstrate that spot swapping occurs from tissue to background. While this reduces tissue signal, thereby reducing the power of the experiment” . However, the visualization of UMI counts on the tissue coordinates does not imply that the occurrence of UMIs in the background actually reduces the signal in the tissue, thus it would be good if the authors can support this claim in greater detail.

Thanks. As you know, in an ideal experiment, UMI counts at a given tissue spot quantify gene-specific expression at that spot, and background spots show UMI counts of zero since these spots are free of tissue. When spot swapping occurs from tissue to background, some mRNAs expressed at a given spot in the tissue are not measured at that spot (rather, they leave the spot and are bound by background probes). This effectively reduces the UMI counts for some genes at some tissue spots. Any tests using UMI count data for a gene affected by spot swapping from tissue to background will have a smaller sample size (compared to the ideal case where bleeding from the tissue to background did not occur). The smaller sample size will result in lower power, all other things being equal.

In an effort to clarify the text, we have changed “signal” to “expression levels”: “While this reduces expression levels at tissue spots, thereby reducing the power of the experiment, a bigger concern is spot swapping from one tissue spot to another, as this confounds downstream analyses.”

4) In Figure 2 the authors evaluate spot-swapping events in chimeric tissue. This represents an interesting approach to test for spot-swapping/ mRNA diffusion in ST experiments. However, the human tissue analyzed by the authors seems to be surrounded entirely by mouse tissue, less a small part at the top of the human tissue. Therefore, it would be helpful to know how background spots, especially with regards to the human tissue in the chimeric experiment, were selected given that most of the background spots (spots adjacent to the human tissue) actually represent mouse tissue. Background spot selection does not become entirely clear as spot-swapping events are reported in proportions (Figure 2b).

Thanks. We regret that this was not more clear. In the human/mouse chimeric experiments, three sets of spots are annotated: human spots, mouse spots, and mixture spots (containing both human and mouse cells). Figure 2a shows these annotations for the experiment that you are referring to, where most of the human spots are surrounded by mouse spots. The full slide (with the background spots included) is shown in Figure 2c. The background spots are those spots outside the tissue region (not annotated as human, mouse, or mixture). These spots

include the ones that you have noted (a small section at the top) along with all other spots on the slide that are not annotated as human, mouse, or mixture. In Figure 2c, we use white to mark the boundary separating tissue and background; mixture spots are also shown in white. To make this more clear, we have added a note to the legend of Figure 2: “Tissue spots on the perimeter as well as spots annotated as mixtures were removed prior to calculating the proportions in panel (b) in an effort to ensure that the effects shown are not due to spots on the tissue-background boundary.”

We have also added the following to the Methods section on Human and mouse tissue spot annotation in the chimeric experiment: “Background spots are defined as those spots on the slide outside the tissue region (not annotated as human, mouse, or mixture).”

5) Between lines 151 - 155 the authors describe the performance evaluations on simulated and case data.

a) While they describe five different simulations in the methods section (from line 519), they only display the results of four simulations in the referenced supplementary tables, with SimV missing in the text and tables.

SimV results were conducted to evaluate SoupX and DecontX, and in particular their performance in downstream analyses to identify spatially variable genes. Results are shown (and compared with SpotClean) in Supplementary Figures 11 and 12, with a discussion given in Supplementary Section S1.

b) In addition, it is confusing why the results of each simulation differ, if the same dataset was taken to perform the simulations for decontamination. If a subset of expression values is used as input by each simulation it would be more reproducible to use the same seed. This way the results of the comparison of the different SpotClean simulations and the existing decontamination methods (SoupX and DecontX) could be presented altogether.

The difference among SimI - IV is the way in which contamination is simulated. SpotClean models spot swapping using a mixture of a Gaussian kernel and a global kernel. SimI generated data following this assumption (i.e. using Gaussian kernel). To test the extent to which SpotClean is robust to departures from this assumption, we simulated data under different contamination scenarios. Specifically, in SimII, SimIII, and SimIV, the contamination weights are given by a Linear, Laplace, and Cauchy kernel, respectively. This was noted in Methods under the Simulation Section. However, to stress this important point, we have revised this section in Methods to read as follows:

“Additional simulations are similar, but for SimII, SimIII, and SimIV the proximal contamination weights are given by a Linear, Laplace, and Cauchy kernel, respectively. This allows us to

investigate the extent to which SpotClean is robust to departures from the Gaussian kernel assumption.”

The results shown in Supplementary Tables 2-5 suggest that SpotClean is robust to a variety of forms of contamination. Specifically, in spite of the different forms of contamination, results are relatively stable. The SpotClean MSE averaged across simulated replicates for the LIBD_151507 simulation, for example, is 15.04, 14.96, 14.998, and 15.39 for Sims I-IV, respectively; for the human_spinalcord based simulation, the average MSE is 13.19, 14.43, 13.41, and 13.15 for the four simulation scenarios, respectively.

Finally, we note that all of our simulations are reproducible as we have controlled random seeds. The source codes are available at our Github site:

https://github.com/zijianni/codes_for_SpotClean_paper

c) Further, it is not clear in the manuscript if the authors compared their method with other existing methods. The statement “Supplementary Tables 2-5 show the mean squared error (MSE) between true and 154 decontaminated gene expression in simulated datasets; SpotClean provides better estimates of 155 expression, reducing the MSE by over 20% in most simulations.” gives the impression that comparisons were only performed between different SpotClean simulations and does not specify in comparison to which decontamination methods/simulations the MSE is reduced by over 20%. In my opinion - it would be important to elaborate and clarify this statement as the authors are directly comparing the performance of their model with published and peer reviewed methods.

The comparison in this section is between data not decontaminated by SpotClean vs. SpotClean decontaminated data. Since this is the first time spot swapping has been identified and defined, there are no comparable methods.

We note that SoupX and DecontX are similar methods in the sense that they adjust for contamination in single-cell RNA-seq experiments. However, they are not appropriate in this setting given that they adjust for a different type of contamination (namely, ambient RNA in droplet based experiments) and they do not accommodate the spatial nature inherent to spot swapping. Consequently, they perform poorly in the spatial transcriptomics setting. While we discuss this in detail in Supplementary Section S1, we did not think it was appropriate to show results from methods not designed for this purpose (especially when the results are, not surprisingly, quite poor).

6) The authors perform simulations in order to test whether their model outperforms existing methods for removing contaminations of ambient RNA in single cell datasets (SoupX and DecontX), in the spatial context this may underlie different distributions.

a) However, from the manuscript it is understood that the authors assume a Poisson distribution of uncontaminated counts, thus it would be good to elaborate why multiple simulations were performed.

SpotClean assumes the uncontaminated counts follow a Poisson distribution, as you mentioned. However, the mean of the distribution is parameterized by components that model proximal contamination and distal contamination. The proximal contamination is modeled by a Gaussian kernel which accommodates dependence among nearby spots due to spot swapping. Multiple simulations are performed so that we can evaluate SpotClean under the Gaussian assumption (Sim I) and when there are departures from the Gaussian assumption (Sim II, III, and IV assume linear, Laplace, and Cauchy, respectively).

b) Further, the authors compared the decontamination methods on spatial data. It would be interesting to see how SpotClean performs on scSeq data, for which the other two algorithms (SoupX and DecontX) were designed, in order to potentially highlight the strength of SpotClean in the spatial context in comparison to other decontamination methods. To this end the authors should attempt to run SpotClean on publicly available single cell data and compare the performance to the referenced SoupX and DecontX.

As we mentioned above, we compared SpotClean with SoupX and DecontX to stress the fact that methods designed for decontamination in single-cell RNA-seq experiments are not appropriate in the spatial transcriptomics setting given that they do not accommodate the spatial nature of spot swapping. SpotClean was not developed for single-cell RNA-seq data and, not only is it not appropriate in the single-cell setting, it is not possible to apply it to single-cell data.

To provide more detail, spot swapping is an artifact unique to spatial transcriptomics data, and it differs in a number of ways from the contamination in single-cell RNA-seq data that SoupX and DecontX were designed for (so-called ambient RNA contamination). As we mentioned in a previous response, Supplementary Section S1 discusses this in detail. In short, a major difference between the contamination in spatial transcriptomics data (spot swapping) and single-cell RNA-seq data (ambient RNA contamination) is that spot swapping is a spatial effect whereas ambient RNA contamination is assumed to be randomly distributed across all cells. We compared SpotClean with SoupX and DecontX because SoupX and DecontX are decontamination methods and they can be applied to spatial transcriptomics data given that the only required input is a gene-by-barcode matrix (barcode here can represent cells in single-cell RNA-seq data or spots in spatial transcriptomics data) .

Unfortunately, SpotClean is unable to apply to single-cell RNAseq data since SpotClean relies on spatial location (i.e. the physical position of the spots) to model and adjust for spot swapping. It also requires both tissue spots and background spots. As you know, both spatial location information and background spots are unavailable in single-cell RNA-seq data, and so SpotClean is not applicable.

7) If understood correctly, SpotClean-decontamination is performed before clustering/factorization and differential gene expression analysis (DGEA) for clusters. The authors report that marker genes within breast cancer samples show higher fold changes and significance (p-values) in the respective clusters (line 167 -172).

a) Do these observations also have an impact on the general number of genes/ gene composition of the individual clusters and if so how is the general clustering affected? Are for example more or less clusters identified after decontamination, inevitably changing the composition of some clusters and how does this impact the biological interpretation of the data?

Thank you for this question. This is an important point that we had discussed in our R package vignette:

<https://github.com/zijianni/SpotClean/blob/5ad83622459bf4e35f7c8624d1cb8b4a02650071/vignettes/SpotClean.Rmd#L99>

“SpotClean will not alter clusters substantially in most datasets given that clusters are largely determined by relatively few highly expressed genes. While clusters may become slightly better defined, in most cases we do not see big differences in the number of clusters and/or relationships among clusters after applying SpotClean.”

To address your question in more detail, we considered the 15 datasets analyzed in the manuscript. For each dataset, we estimated the number of clusters in the data using the clustering pipeline in Seurat, where the number of clusters need not be pre-specified. Note that the parameter, *resolution*, controls the relative number of clusters; the default value was used for each dataset. As shown below, the number of clusters stays the same, or changes by one, for most datasets.

Number of clusters before and after decontamination for 15 datasets:

Dataset	#clusters in raw	#clusters in SpotClean
LIBD_151507	6	7
LIBD_151508	6	6
LIBD_151669	8	7
LIBD_151670	8	7
LIBD_151673	7	9
LIBD_151674	9	8
Visium_FFPE_Human_Breast_Cancer	13	12

V1_Human_Lymph_Node	12	12
Targeted_Visium_Human_SpinalCord_Neuroscience	6	7
V1_Breast_Cancer_Block_A_Section_2	15	14
V1_Adult_Mouse_Brain	19	17
V1_Mouse_Kidney	10	11
HM-1	9	10
HM-2	9	9
HM-3	6	6

As we noted in our discussion, “Since the probability of a spot-specific barcode binding reads from another spot increases as the distance between spots decreases, most of the adjustments made by SpotClean are local (i.e. reads are reassigned from one spot to a nearby spot). Given this, SpotClean will have only a modest impact on some downstream analyses, but a more major impact on others. Specifically, since average expression within a region will remain largely unchanged post SpotClean, downstream analyses that rely on average expression (e.g. DE analyses) will show only slight improvements over the raw data, as shown here. Modest improvements can also be expected for data where clusters are easily separated. However, for more specific analyses and/or more subtle signals, the effects of SpotClean are greater. Specifically, SpotClean provides substantial improvements in marker gene analyses by decreasing expression in regions where markers are known to be lowly expressed, while maintaining expression levels in other regions. In addition, SpotClean substantially improves clustering results and spot annotations in situations where regions are not easily separated, which may have important implications for clinical applications of the ST technology (e.g. in cancer diagnosis and staging).”

An example of this last point was given in Figure 6, where we estimate malignancy burden in a breast cancer study. A new example related to clustering is detailed in response to 8b below, where we show results from a colorectal dataset where the number of clusters is the same in raw and SpotClean decontaminated data, but the composition changes after SpotClean.

b) The authors mention generating ARI scores to compare the clustering results but it does not become entirely clear what the input for the ARI is. I am assuming the authors are comparing malignant spot annotations from visual annotations (of the H&E image) and the annotations of malignant cancer spots based on transcription. It would be essential to elaborate what exactly was compared for the ARIs.

Thanks! You are correct, and we have added a bit more detail to the Methods section on Clustering and ARI in an effort to make this more clear. This section now reads: “For each

cancer case study analysis (breast, pancreatic, and colorectal), the Seurat pipeline was applied under default settings to the raw and decontaminated data to produce UMAP plots. For the breast cancer data and the pancreatic cancer data, tumor spots were clustered using k-means clustering (k=2) of the top 50 PCs calculated in the Seurat pipeline. For the colorectal cancer data, tumor spots were clustered using BayesSpace under default settings. In the H&E image, tissue spots were labelled as tumor and non-tumor based on visual inspection. The adjusted rand indexes (ARI) were calculated between cluster labels and tumor/non-tumor labels.”

c) Further the ARIs presented in Figure 7 d are both very low, indicating close to random assignment of spots to clusters (In this case malignant and non-malignant). Further the difference between the ARIs only has a magnitude of 0.05) which does not appear very convincing in that the decontaminated data is assigning spots notably better to the expected malignant cluster. This observation also applies to Figure 5, where the general ARIs are more convincing (0.86 and 0.89) but the difference between the ARIs is even lower (0.03). In the case of Figure 4 ARI values are most convincing but it might be advisable to consider an alternative/additional measure to compare clustering/correct spot assignment performance.

Yes, we see your point. While the ARIs suggest that SpotClean decontaminated data results in more specific clusters, the improvements are not dramatic in most cases, and the overall ARI in Figure 7 is quite low in both raw and SpotClean decontaminated data. As we mentioned in response to your question 7a, we had noted in our vignette and in the discussion of the manuscript that since clusters are largely determined by relatively few highly expressed genes, SpotClean will not alter clusters substantially in most datasets, but most often clusters will become better defined.

In Figures 4 and 5, there seems to be relatively good discriminatory power to identify differences between two groups (e.g. tumor vs. non-tumor). In these cases, the ARIs are relatively high. There is less power to distinguish among more subtle groupings (e.g. multiple tumor subpopulations in the original Figure 7). This is reflected in the ARIs. However, in all cases (whether the ARIs are relatively high, or not), SpotClean results in improved clusters. In the case of Figure 4, the ARI increased from 0.71 to 0.80 following SpotClean, a 13% increase. In Figure 5, the increase is smaller (0.86 to 0.89, 3%), but of course there is less room for improvement in cases where the groups are well differentiated in raw data (i.e. when the clustering results are good to start with). In the last case (Figure 7), there was a 36% increase in ARI following SpotClean, although we acknowledge that the ARIs are relatively low in both the raw and SpotClean decontaminated data (0.14 in the raw and 0.19 in the SpotClean decontaminated data). As you suggested, we have removed this plot and instead now show improvements in clustering as assessed by marker composition and malignancy scores (new Figure 7c and d in the manuscript; with details given in response to 8b, below).

d) More generally, it is questionable if using ARI to enable quantifying the improved separation of the tumor and non-tumor regions, as stated between line 169 and line 171 as well as line 272 – 273 is an accurate strategy. An ARI is a comparative measure to

evaluate if two cluster results are similar to each other. Based on the above described concern about the low magnitudes between ARI values, the authors should justify the use of ARIs to quantify the improved clustering results further.

As you mentioned, ARI is used to quantify similarity between two partitions. Here, the partitions are tumor vs. non-tumor as annotated by clusters and tumor vs. non-tumor as annotated by H&E spot annotations. The ARIs are used here since we wanted to quantify how similar the clusters following SpotClean (or not) compare with the gold-standard annotation from the H&E image, the idea being that if SpotClean is improving the quality of the data, then the tumor/non-tumor clusters following SpotClean should have improved agreement with the image annotations. Having said that, we acknowledge that the discriminatory power for the last dataset is low as reflected in relatively low ARIs for both the raw and SpotClean decontaminated data. As mentioned in our response to 7c, we have removed this evaluation.

For reasons given in 7c, we do think it's useful to use ARI to quantify improvements in clusters and have left them in Figures 4 and 5. Other metrics for quantifying similarity between two partitions show similar results. Specifically, we considered normalized mutual information (NMI), and the Fowlkes-Mallows index (FMI). The values in the raw vs. SpotClean decontaminated data are shown below.

Breast Cancer (Figure 4d):

ARI: 0.71 vs 0.80, NMI: 0.58 vs 0.67, FMI: 0.88 vs 0.92

Pancreatic Cancer (Figure 5c):

ARI: 0.86 vs 0.89, NMI: 0.76 vs 0.80, FMI: 0.95 vs 0.96

Colorectal Cancer (original Figure 7d):

ARI: 0.14 vs 0.19, NMI: 0.13 vs 0.17, FMI: 0.58 vs 0.61

8) In Figure 7 the authors show the effect of decontamination on clustering results of raw Visium spatial data. Panel c shows the assignment of 6 clusters, including 5 tumor clusters for the raw data but only 4 tumor clusters for the decontaminated data. How do these clusters differ in their total gene expression profiles?

A detailed investigation of the differences among some of the tumor clusters can be found in our response to 8b, below.

Differences between tumors within the same tissue are another essential quality (apart from the position of the tumor) to ensure the most optimal treatment.

a) Based on their claim that "SpotClean decontaminated data leads to improved delineation of tumor and non-tumor regions and improved identification of clusters.", how are the authors able to support their claim that the clustering performed after

decontamination using SPOTClean resembles the actual tumor biology better than the raw data? The evidence needs to be explained more explicitly.

This claim that SpotClean decontaminated data leads to improved delineation of tumor and non-tumor regions is based on the fact that the SpotClean decontaminated data gives tumor vs. non-tumor clusters that more closely align with the gold-standard H&E tumor vs. non-tumor as quantified by ARI. We noted this when discussing results shown in Figures 4 and 5 (lines 170-172 of the original manuscript): Specifically, we noted that SpotClean “leads to improved separation of the tumor and non-tumor regions via clustering, as shown visually, and quantified by ARI scores. Similar results are shown in Figure 5 in a study of pancreatic cancer.”

The sentence you’re referring to is in discussing clustering results from Figure 7 which we have since removed since we are no longer showing ARI for Figure 7, but the same evidence (ARI improvement) was what we were referring to in the original manuscript. In addition to ARI, in Figure 7b, we show malignancy composition scores estimated via SPOTlight. As shown, the SpotClean decontaminated data has higher malignancy scores in tumor regions, and lower scores in non-tumor regions. Finally, in Figure 7c, we had highlighted two regions where the SpotClean decontaminated tumor clusters more closely resembled the tumor annotations in the H&E stain. This panel has been replaced (see response to 8b).

b) Further validation of differences between the tumor clusters would be necessary, this could for example be done by the analysis of markers for certain tumor subtypes or a more detailed description of the biological differences between the identified tumor clusters.

To be clear, we didn’t make claims regarding differences between tumor clusters or tumor subtypes, as detailed information about tumor subpopulations is not provided in the datasets we considered. Rather, we focused on delineation between tumor and non-tumor spots since we have a reasonable gold standard in the H&E stain. Having said that, we do agree that a more detailed analysis of tumor clusters could provide additional support for SpotClean. Toward this end, in the colorectal cancer case study, we used BayesSpace clustering to cluster tumor spots in the raw and in the SpotClean decontaminated data. BayesSpace requires a user to specify the number of clusters, which we fixed at 6 so that we could focus directly on how cluster composition changes (even if number does not). With both the raw and SpotClean decontaminated data clustered into 6 clusters, we see differences among the clusters. The main difference is that SpotClean identifies a new cluster (SpotClean tumor cluster 1); see the new Figure 7 shown below.

To investigate the novel cluster identified by SpotClean tumor cluster 1, we compared this group with all other tumor spots. If SpotClean tumor cluster 1 is in fact a distinct group of spots, we’d expect to see biologically relevant DE genes. Using Seurat’s pipeline with default settings for identifying DE genes, 74 genes were identified as DE at adjusted p-value ≤ 0.01 . Nine of the top 10 DE genes are immunoglobulin genes, suggesting that this group of spots is a mixture of tumor cells and normal tumor-infiltrating immune cells. Multiple analyses further suggest that

this is the case. First, a careful look at the H&E stain shows that this group of spots is non-normal, but distinct from other tumor regions. Second, strong expression of immunoglobulin genes is largely specific to SpotClean cluster 1. Finally, the average malignancy score for this group is lower than other tumor clusters, but higher than normal spots, further suggesting that this group of spots contains both tumor and tumor-infiltrating immune cells. Taken together, this evidence suggests that the novel cluster identified by SpotClean maintains biologically relevant information and, in this case, provides for a more specific clustering. The updated Figure7 is shown below; and we have modified the manuscript text to read as follows (lines 191-205):

“Similar results are shown in Figure 7 in a study of colorectal cancer where SpotClean decontaminated data leads to improved delineation of tumor and non-tumor regions as evidenced by enhanced tumor malignancy scores in tumor spots, and lower malignancy scores in non-tumor spots, compared with raw data (Figure 7b). SpotClean also identifies a novel cluster (SpotClean tumor cluster 1 shown in Figure 7c); and multiple analyses suggest that this cluster is a distinct tumor sub-population containing both tumor and tumor-infiltrating immune cells. First, a careful look at the H&E stain shows that this group of spots is non-normal, but distinct from other tumor regions (red boxes, Figure 7a). Second, 9 of the top 10 genes identified as DE between SpotClean tumor cluster 1 and other tumor clusters are immunoglobulin marker genes (from 74 total DE genes with adjusted p-value ≤ 0.01); and immunoglobulin expression for these 9 genes is largely specific to this cluster (Figure 7d). Finally, the average malignancy score for this group is lower than other tumor clusters, but higher than normal spots, further suggesting that this group of spots contains both tumor cells and tumor-infiltrating immune cells (average malignancy scores at normal, tumor cluster 1, and other tumor spots are 0.384, 0.430, and 0.477, respectively). Taken together, this evidence suggests that the novel cluster identified by SpotClean maintains biologically relevant information and, in this case, provides for a more specific clustering that captures subtle structure present in the tissue.”

Figure 7: Data from a study of human colorectal cancer, sample human_colorectal. Panel (a) shows the H&E image (left) and spots annotated as tumor vs. non-tumor via a pathologist's visual inspection (right). Red boxes highlight the spots belonging to SpotClean's tumor cluster 1 (panel (c)). (b) Malignant spot composition as estimated via SPOTlight¹⁴ is shown for the raw (left) and SpotClean decontaminated data (right). SpotClean results in higher malignancy scores in tumor regions, and lower in normal regions. (c) BayesSpace¹⁵ clustering for the raw data (top) and SpotClean decontaminated data (bottom). The SpotClean decontaminated data identifies a novel cluster (SpotClean tumor_1, red boxes). The SpotClean tumor_1 spots are distinct on the H&E stain (red boxes in panel (a)) and likely contain tumor-infiltrating immune cells as evidenced by high expression in the immunoglobulin markers shown in panel (d).

9) The authors convincingly demonstrate that the method they developed is able to reduce noise signals in spatial data. This is mainly presented by spatial expression profiles of known spatial marker genes (Figure 3,4 and additional Supplementary Figures). However, as the method is removing noise and increased fold-changes and smaller p-values for DEGs, it would be interesting to evaluate whether the method finds additional DEGs between spatial clusters, or if some genes that are considered differentially expressed before decontamination disappear after decontamination.

SpotClean does find additional DE genes, as well as genes that were DE but become EE post SpotClean. We did not discuss these since there is no gold-standard to which we can compare results. Specifically, we do not have complete lists of DE genes; we simply have a list of some DE genes that have been reported, and in some cases independently validated, in previous studies. For those, as you mentioned, we see improved fold-changes and p-values.

We did find some interesting results in the case studies. For example, in the breast cancer case study, SpotClean finds 132 new DE genes with FDR controlled at 5%. Many of these are known to be involved in breast cancer. The top 5 are ACTG1, GAPDH, TPT1, FN1, and CFL1. However, these 5 appear on the raw data's DE list if we relax FDR to 14%. In short, since SpotClean does not change the data in major ways, there is considerable overlap of the DE genes and it is difficult to make qualitative statements about general performance without a gold standard list.

10) From the experimental perspective, permeabilization efficiency plays a crucial role for the success of a Visium or Spatial Transcriptomics experiment. This permeabilization needs to be optimized for different tissue types and over-permeabilization can lead to higher diffusion rates between spots, i.e. spot-swapping. Have the authors observed substantial differences of bleeding rates $r\beta$ and contamination rates $r\gamma$ between individual tissue sections and/or tissue types. This could, in addition to the suggested improvements of the clustering and DGEA results, serve as a valuable quality control measurement for the resulting sequencing data and increase the relevance of the suggested method.

Great idea - thanks.

To address this question, ideally we'd like a series of Visium experiments where only permeabilization times vary. Unfortunately, as you know, such experiments are unavailable. To get some insight using existing data, we compared results from samples with different permeabilization times in different studies. Given that most of the Visium data at the 10x Genomics website does not contain permeabilization information, we collected additional data

from recent publications using Visium with known permeabilization times. The datasets we used are listed below:

6min permeabilization: GEO GSM5808054, GSM5808055, GSM5808056, GSM5808057

12min permeabilization: Our chimeric samples HM-1, HM-2, HM-3

18min permeabilization: SpatialLIBD data, GSM5691526, GSM5691527, GSM5691528, GSM5691529

30min permeabilization: GSM5388414, GSM5388415, GSM5213483, GSM5213484, GSM5726156, GSM5726157

Dataset	Permeabilization time	Bleeding rate r_β	Distal rate r_γ
GSM5808054	6min	0.55	0.67
GSM5808055	6min	0.54	0.46
GSM5808056	6min	0.56	0.14
GSM5808057	6min	0.58	0.51
HM-1	12min	0.47	0.2
HM-2	12min	0.46	0.16
HM-3	12min	0.4	0.34
GSM5691526	18min	0.73	0.67
GSM5691527	18min	0.57	0.45
GSM5691528	18min	0.6	0.23
GSM5691529	18min	0.71	0.31
LIBD_151507	18min	0.37	0.63
LIBD_151508	18min	0.37	0.55
LIBD_151669	18min	0.41	0.42
LIBD_151670	18min	0.38	0.46
LIBD_151673	18min	0.4	0.44
LIBD_151674	18min	0.43	0.43
DD073R_A1	30min	0.47	0.26
DD073R_D1	30min	0.31	0.27
GSM5213483	30min	0.44	0.95
GSM5213484	30min	0.43	0.53
GSM5726156	30min	0.49	0.17
GSM5726157	30min	0.48	0.32

As shown in the table and figures above, there are no clear patterns between permeabilization times and estimated bleeding rates (r_β) or distal contamination rates (r_γ). Datasets coming from the same study tend to have similar bleeding rates, but we do not observe a clear trend of bleeding rates as a function of permeabilization time. Of course here the estimated effect of permeabilization is confounded by different species, tissue types, sample preparations, etc. As more data become available, it will be interesting to continue to assess whether or not there is a relationship between permeabilization time and bleeding rates as we agree that this could be another useful metric provided by SpotClean.

11) From line 351 - line 375, the authors describe the SpotClean method in detail. They describe variable G to be a set of genes. It would be helpful to elaborate how this set of genes G is determined, i.e. how are the genes selected, how large is the set of genes, etc. - are these based on e.g. a specific number of variable genes in the count data?

We agree that this information would be useful when SpotClean is first introduced. As stated in Methods (original manuscript line 549), we noted that “SpotClean decontaminates genes with average expression above 1, high variance as determined by Seurat's FindVariableFeatures() function, or both.” We now provide some information in the section in the main text where SpotClean is first introduced, and we point the reader to Methods. In particular, the SpotClean section in the text now reads: “Let K be the total number of spots, G be the set of genes, I_t be the set of tissue spots with cardinality $|I_t|=K_t$, and I_b be the set of background spots with cardinality $|I_b|=K_b$ where $K_t+K_b=K$. As detailed in Methods, G defaults to genes that are highly expressed, highly variable, or both; this default can be relaxed by a user. The true (i.e., uncontaminated) UMI counts are given by...”

We also have more discussion about this important point in our R package vignette:

<https://github.com/zijianni/SpotClean/blob/5ad83622459bf4e35f7c8624d1cb8b4a02650071/vignettes/SpotClean.Rmd#L91>

“Lowly-expressed genes typically contain relatively less information and relatively more noise than highly-expressed genes. SpotClean by default only keeps highly-expressed or highly-variable genes for decontamination (or both). It can be forced to run on manually-specified

lowly-expressed genes. However, even in this case, expression for the lowly-expressed genes is typically not changed very much. Given the high sparsity in most lowly expressed genes, there is not enough information available to confidently reassign UMIs in most cases. However, we do not filter genes by sparsity because there can be interesting genes highly concentrated in a small tissue region. In cases like this, SpotClean is effective at adjusting for spot swapping in these regions. If the defaults are not appropriate, users can either adjust the expression cutoffs or manually specify genes to decontaminate.”

Reviewer #2

This is a very interesting work that delivers an important message for the spatial transcriptome field: the signal you get from one spot is a weighted average of nearby spots. The authors have used multiple publicly available datasets and their own data (by combining human and mouse samples into one slide) to demonstrate this. They have also developed a computational method to correct such signal mixing and demonstrate its effectiveness. Overall, this is an important work that will have impact. In some situations, such as downstream analyses relying on average expression (as pointed out by the authors), such correction of spot mixture may not be crucial, but it is still important to know this when interpreting the results.

Thanks!

I have some minor comments on the method details.

1. For the lower bound on the proportion of spot-swapped reads (LPSS), the estimate from the chimeric experiments was 10-15%. Does it mean that 10-15% of UMI from a spot is from nearby spots? It would be helpful to clarify how is this being calculated. Since this is a lower bound and it may not be unreasonable to believe that twice amount of swap happen (e.g., human to human or mouse to mouse), which gives 20-30% of UMI per spot are from nearby spots. Is that right?

The LPSS reported in Supplementary Table 1 is the overall LPSS. Specifically, $LPSS = (\text{all human UMIs in all mouse spots} + \text{all mouse UMIs in all human spots}) / \text{all UMIs in all tissue spots}$. Consequently, LPSS does not directly quantify how much spot swapping there is at any given spot. It's an estimate of the proportion of spot swapped reads across the entire experiment; and it's a lower bound on that proportion, as you noted, since it does not account for UMIs that might move within species (e.g. human UMIs expressed at one spot that bind probes at another human spot). Given this, we agree that it is not unreasonable to estimate the total percentage of spot-swapped reads to be 20-30%.

Is this estimate consistent with other data?

In short, yes. Since LPSS can only be estimated in the chimeric experiment, to estimate a lower bound on the proportion of spot-swapped reads in typical (non-chimeric) experiments, we consider the percentage of UMIs in background spots. This percentage ranges from 4.2% to 26.9% in the 14 datasets considered (shown in Supplementary Table 1). It is important to note that these percentages depend on the number of background spots (as the number of background spots decreases, the percentage of counts in the background will typically decrease - and if there are no background spots, the percentage is necessarily zero). The chimeric experiments have the highest percentages (20.8%, 17.6%, and 26.9% for HM-1, HM-2, and HM-3, respectively), but they also have the largest number of background spots. Consequently, to make these values comparable across experiments, we considered the normalized proportion of UMIs in background spots, where the percentage is normalized by the number of background spots. When normalized, the results reported for the chimeric experiments are comparable with other experiments. Given this, it is reasonable to assume that the estimates shown are comparable across experiments.

2. What is the identifiability requirement of the model? or in other words, what are required to obtain reliable estimates. For example, if all the spots are homogeneous (but there are still backgrounds), could the SpotClean method work?

Background spots are crucial for SpotClean to model spot swapping and to estimate the bleeding rates and other unknown parameters in the model. We had some discussion about identifiability in our R package vignette:

<https://github.com/zijianni/SpotClean/blob/5ad83622459bf4e35f7c8624d1cb8b4a02650071/vignettes/SpotClean.Rmd#L89>

“While the observed data is a single matrix with a fixed number of columns (spots), the number of unknown parameters is proportional to the number of tissue spots. In the extreme case where all spots are covered by tissue, we have more unknown parameters than observed data values. In this case the contaminated expressions are confounded with true expressions, and SpotClean estimation becomes unreliable. We recommend that the input data have at least 25% spots not occupied by tissue, so that SpotClean has enough information from background spots to estimate contamination.” We have now added this information to the Methods section in a new subsection entitled Minimum number of background spots required for parameter estimation.

To address your question on homogeneity, in a typical ST dataset, UMI counts in the background decrease with increased distance from the tissue. As a result, we don't expect background spots to be homogeneous and we do not have restrictions on the extent of homogeneity in tissue spots.

3. At line 368, " r_{distal} is a distal and $1 - r_{\text{proximal}}$ is a proximal contamination rate”, the distal means a constant bleeding to all the K spots? It is hard to imagine that one spot can bleed somewhere faraway?

Yes, the distal contamination corresponds to a global constant kernel. It's not where we started, but we observed that a single local kernel (Gaussian, t , or other similar kernels) was not able to fully model the spot swapping distribution. In particular, when simulating data from a model without a global kernel, we were unable to capture the extent of UMI counts in background spots. The figure below shows one example for Camk2a in the mouse_brain dataset, a well-known marker in brain tissue involved in signal transduction. The left panel shows the observed expression of Camk2a in the mouse_brain data. The middle panel shows the simulated expression of Camk2a, where the spot swapping effect was simulated using a combination of local and global kernels (this simulation setting is the same as our Siml described in the manuscript). The right panel shows the simulated expression of Camk2a, where the spot swapping effect was simulated using only a local Gaussian kernel. As you can see, the simulated expression using a combination of local and global kernels better resembles the observed data compared to the simulation using the local kernel only. Consequently, we used a weighted sum of a local and global kernel, where the weight r_gamma is learned from data, in order to better approximate the true underlying spot swapping distribution.

4. Parameters are estimated by minimizing residual sum squares. It is reasonable, but somewhat unusual for a Poisson model since variance increases with mean. Is it more appropriate to use Poisson likelihood as objective function?

The residual sum squares (RSS) was used only for estimating the bleeding rate (r_beta) and the distal contamination rate (r_gamma). The underlying true expressions were estimated by maximizing the data likelihood using the EM algorithm. Details can be found in Supplementary Section S3.

It's a good point to ask why they are not estimated together by maximizing the data likelihood. The main reason is that convergence is an issue. Below is a model we developed early on where the bleeding rate and the distal contamination rate were spot-specific parameters, and were estimated together with underlying true expressions in the EM model (note that the notation used below is slightly different than what is used in the manuscript). Due to the large number of unknown parameters as well as the structure of the likelihood (especially the product

of underlying true expressions (μ_{gt}), spot-specific bleeding rate (r_t), and spot-specific distal contamination rate (c_t), the parameters typically did not converge even after 100 iterations.

Our current model is a simplified version of the original model that assigns the bleeding rate and distal contamination rate to be global parameters among spots, and pre-estimates these rates. By doing so, we are able to significantly improve the convergence speed and reduce computation. We also carefully designed the way of choosing initial values to avoid local optima as much as possible (line 385 and line 394 of the original manuscript).

For Review: an Early SpotClean Model with Spot-specific Bleeding Rates and Distal (Global) Contamination Rates

Zijian Ni

March 12, 2022

1 Model Construction

We model contamination of all genes and spots simultaneously and estimates unknown parameters via an EM algorithm. Let K be the total number of spots, G be the set of genes, I_t be the set of tissue spots with cardinality $|I_t| = K_t$, and I_b be the set of background spots with cardinality $|I_b| = K_b$. $K_t + K_b = K$. Observed UMI counts are $\mathcal{D} = \{X_{g,j}\}_{g \in G, j \in I_t \cup I_b}$. Let $\{Y_{g,t}\}_{g \in G, t \in I_t}$ be the unknown true UMI counts in tissue spots before contamination happens. Decompose $Y_{g,t}$ to be

$$Y_{g,t} = S_{g,t} + B_{g,t}$$

where $S_{g,t}$ is the amount of UMI counts staying in tissue spot t , and $B_{g,t}$ is the amount of UMI counts bleeding out. It is convenient to set $Y_{g,b} = S_{g,b} = B_{g,b} = 0$ for background spots $b \in I_b$ since background spots do not express anything. We further decompose $B_{g,t}$ as

$$B_{g,t} = \sum_{j \in I_t \cup I_b} (C_{g,t,j} + N_{g,t,j})$$

where $C_{g,t,j}$ is the amount of global contamination and $N_{g,t,j}$ is the amount of local contamination from tissue spot t to any spot j . The expected values of $C_{g,t,j}$ are constant for all j , whereas the expected values of $N_{g,t,j}$ depend on the distance between t and j . As a result,

$$R_{g,j} = \sum_{t \in I_t} (C_{g,t,j} + N_{g,t,j})$$

is the amount of contaminated UMIs spot j received due to spot swapping, and

$$X_{g,j} = S_{g,j} + R_{g,j}$$

is the observed data for any gene g and any spot j . We assume all the variables of UMI counts are Poisson distributed and independent among genes and spots. Specifically,

$$\begin{aligned} Y_{g,t} &\sim \text{Poisson}(\mu_{g,t}) \\ X_{g,j} &\sim \text{Poisson}(\eta_{g,j}) \end{aligned}$$

We define the weight between tissue spot t to any spot j based on the Gaussian kernel

$$w_{t,j} = \frac{\exp\left\{\frac{d_{t,j}^2}{-2\sigma_g^2}\right\}}{\sum_{j' \in I_t \cup I_b} \exp\left\{\frac{d_{t,j'}^2}{-2\sigma_g^2}\right\}}$$

where $d_{t,j}$ is the physical Euclidean distance between spot t and j in pixels, and σ_g is the contamination radius. A small contamination radius implies the local contamination is more concentrated around the tissue spot. The estimation of σ_g will be pre-specified proportional to the expression level of each gene to avoid overparameterization and computational challenges.

2 Model Parameterization and Estimation

Let $\{r_t\}_{t \in I_t}$ be the bleeding rates and $\{c_t\}_{t \in I_t}$ be the global contamination rates in tissue spots. Following similar ideas to those in Method I, we can get

$$\begin{aligned} S_{g,t} &\sim \text{Poisson}(\mu_{g,t}(1 - r_t)) \\ C_{g,t,j} &\sim \text{Poisson}\left(\mu_{g,t} r_t c_t \frac{1}{K}\right) \\ N_{g,t,j} &\sim \text{Poisson}(\mu_{g,t} r_t (1 - c_t) w_{t,j}) \end{aligned}$$

As a result,

$$\eta_{g,j} = E(X_{g,j}) = \begin{cases} \sum_{t \in I_t} \mu_{g,t} r_t \left[c_t \frac{1}{K} + (1 - c_t) w_{t,j} \right], & \text{if } j \in I_b \\ \mu_{g,j} (1 - r_j) + \sum_{t \in I_t} \mu_{g,t} r_t \left[c_t \frac{1}{K} + (1 - c_t) w_{t,j} \right], & \text{if } j \in I_t \end{cases}$$

Parameter Estimation via Expectation-Maximization We estimate the unknown parameters $\{r_t\}_{t \in I_t}$, $\{c_t\}_{t \in I_t}$ and $\{\mu_{g,t}\}_{g \in G, t \in I_t}$ for all genes and tissue spots by maximizing the data likelihood using an Expectation-Maximization (EM) algorithm?. Under the EM framework, the observed data are $\{X_{g,j}\}_{g \in G, j \in I_t \cup I_b}$ with log-likelihood

$$l_D = \sum_{g \in G} \sum_{j \in I_t \cup I_b} \{X_{g,j} \log \eta_{g,j} - \eta_{g,j}\} + \text{constant},$$

and the complete data are $\{S_{g,t}, C_{g,t,j}, N_{g,t,j}\}_{g \in G, t \in I_t, j \in I_t \cup I_b}$ with log-likelihood

$$\begin{aligned} l_C &= \sum_{g \in G} \sum_{t \in I_t} [S_{g,t} \log(\mu_{g,t}(1 - r_t)) - \mu_{g,t}(1 - r_t)] \\ &+ \sum_{g \in G} \sum_{t \in I_t} \sum_{j \in I_t \cup I_b} \left[C_{g,t,j} \log(\mu_{g,t} r_t c_t \frac{1}{K}) - \mu_{g,t} r_t c_t \frac{1}{K} \right] \\ &+ \sum_{g \in G} \sum_{t \in I_t} \sum_{j \in I_t \cup I_b} [N_{g,t,j} \log(\mu_{g,t} r_t (1 - c_t) w_{t,j}) - \mu_{g,t} r_t (1 - c_t) w_{t,j}] + \text{constant} \\ &= \sum_{g \in G} \sum_{t \in I_t} \left(S_{g,t} \log(\mu_{g,t}(1 - r_t)) + \sum_{j \in I_t \cup I_b} \left[C_{g,t,j} \log(\mu_{g,t} r_t c_t \frac{1}{K}) + N_{g,t,j} \log(\mu_{g,t} r_t (1 - c_t) w_{t,j}) \right] - \mu_{g,t} \right) \\ &+ \text{constant} \end{aligned}$$

Let $\{\mu_{g,t}^{(n)}, r_t^{(n)}, c_t^{(n)}\}_{g \in G, t \in I_t}$ be the parameter values at the n -th iteration. The E-step involves computation of the expectation of latent variables conditioning on observed data and parameter values at the current iteration. From the assumption of independence,

$$\begin{aligned} S_{g,t}^{(n)} &:= E[S_{g,t} | \mathcal{D}] = E[S_{g,t} | X_{g,t}] = X_{g,t} \frac{\mu_{g,t}^{(n)} (1 - r_t^{(n)})}{\eta_{g,t}^{(n)}} \\ C_{g,t,j}^{(n)} &:= E[C_{g,t,j} | \mathcal{D}] = E[C_{g,t,j} | X_{g,j}] = X_{g,j} \frac{\mu_{g,t}^{(n)} r_t^{(n)} c_t^{(n)} \frac{1}{K}}{\eta_{g,j}^{(n)}} \\ N_{g,t,j}^{(n)} &:= E[N_{g,t,j} | \mathcal{D}] = E[N_{g,t,j} | X_{g,j}] = X_{g,j} \frac{\mu_{g,t}^{(n)} r_t^{(n)} (1 - c_t^{(n)}) w_{t,j}}{\eta_{g,j}^{(n)}} \end{aligned}$$

The M-step involves maximizing the complete log-likelihood after plugging in the condi-

tional expectations in the E-step:

$$l_C^{(n)} = \sum_{g \in G} \sum_{t \in I_t} \left(S_{g,t}^{(n)} \log(\mu_{g,t}(1 - r_t)) + \sum_{j \in I_t \cup I_b} \left[C_{g,t,j}^{(n)} \log(\mu_{g,t} r_t c_t \frac{1}{K}) + N_{g,t,j}^{(n)} \log(\mu_{g,t} r_t (1 - c_t) w_{tj}) \right] - \mu_{g,t} \right)$$

$$\frac{\partial l_C^{(n)}}{\partial \mu_{g,t}} = \frac{S_{g,t}^{(n)} + \sum_{j \in I_t \cup I_b} (C_{g,t,j}^{(n)} + N_{g,t,j}^{(n)})}{\mu_{g,t}} - 1$$

$$\frac{\partial l_C^{(n)}}{\partial r_t} = \sum_{g \in G} \left[\frac{S_{g,t}^{(n)}}{r_t - 1} + \frac{\sum_{j \in I_t \cup I_b} (C_{g,t,j}^{(n)} + N_{g,t,j}^{(n)})}{r_t} \right]$$

$$\frac{\partial l_C^{(n)}}{\partial c_t} = \sum_{g \in G} \sum_{j \in I_t \cup I_b} \left[\frac{C_{g,t,j}^{(n)}}{c_t} + \frac{N_{g,t,j}^{(n)}}{c_t - 1} \right]$$

$$\mu_{g,t}^{(n+1)} = S_{g,t}^{(n)} + \sum_{j \in I_t \cup I_b} (C_{g,t,j}^{(n)} + N_{g,t,j}^{(n)})$$

$$r_t^{(n+1)} = \frac{\sum_{g \in G} (\mu_{g,t}^{(n+1)} - S_{g,t}^{(n)})}{\sum_{g \in G} \mu_{g,t}^{(n+1)}}$$

$$c_t^{(n+1)} = \frac{\sum_{g \in G} \sum_{j \in I_t \cup I_b} C_{g,t,j}^{(n)}}{\sum_{g \in G} (\mu_{g,t}^{(n+1)} - S_{g,t}^{(n)})}$$

Reviewers' Comments:

Reviewer #1:

Remarks to the Author:

First off, the authors have addressed and answered the comments and concerns very elaborately and improved the manuscript by further increasing the relevance of the study. The authors have provided sufficient responses to the majority of my comments, please see additional questions/comments below (numbers below correlate with the numbers in the previous remarks from this reviewer). Once these remaining comments have been addressed it is my opinion that this manuscript has met the criteria for publication.

2.The authors implemented and visualized the data well, however upon inspecting the graphs, I fail to see the stated higher bleeding rates in mitochondrial and lncRNAs for the three mentioned datasets, specifically (human lymph node; breast cancer; and mouse brain) in the boxplots shown in the rebuttal. I believe that it would make sense to include a simple statistical test to validate more clearly that there is no significant difference between the groups. I further believe that this data would be interesting to include in the supplementary material of this study to serve as a quality control tool for future studies.

3.Thank you for elaborating further on the motivation of this statement. While I agree with the authors that the diffusion of spots decreased the number of UMIs in the original spot (beneath the tissue), I consider the statement that these expression levels reduce the power of the entire experiment to be quite strong. Thus, if the authors want to stand by this comment I believe they need to show if UMI counts are in fact reduced significantly or rather have a large effect size in neighboring spots. One spot generally harbors a high number of UMIs (this is at least expected for highly expressed genes). Statistically, once sample sizes (UMIs) are very high, even small differences become significant, which might not necessarily be of the same biological significance and might therefore not have a meaningful effect on the interpretation of the observations. Alternatively, the authors could simply remove this statement, so it reads: "While this possibly reduces expression levels at tissue spots, a bigger concern is spot swapping from one tissue spot to another, as this confounds downstream analyses."

Suggestion:

* 10.I appreciate the author's effort to address the suggestion and agree with the authors that there is no clear correlation between permeabilization time and bleeding or distal rate in the analyses they present. I would ask the authors to consider including these findings in the supplementary data to first, show that their experiment is not influenced by the quality or permeabilization time of the tissue section and second, to advocate that the research community can use the SpotClean method to calculate bleeding and distal rates (using SpotClean) to potentially explain unexpected or unclear observations in their spatial experiments.

Reviewer #2:

Remarks to the Author:

the authors have addressed all my concerns. Congratulations for this nice work.

We thank the reviewers for their second-round review; their detailed comments have helped to improve the manuscript. Below we provide a point-by-point response, with reviewer comments given in bold.

Reviewer #1 (Remarks to the Author):

First off, the authors have addressed and answered the comments and concerns very elaborately and improved the manuscript by further increasing the relevance of the study. The authors have provided sufficient responses to the majority of my comments, please see additional questions/comments below (numbers below correlate with the numbers in the previous remarks from this reviewer). Once these remaining comments have been addressed it is my opinion that this manuscript has met the criteria for publication.

2. The authors implemented and visualized the data well, however upon inspecting the graphs, I fail to see the stated higher bleeding rates in mitochondrial and lncRNAs for the three mentioned datasets, specifically (human lymph node; breast cancer; and mouse brain) in the boxplots shown in the rebuttal. I believe that it would make sense to include a simple statistical test to validate more clearly that there is no significant difference between the groups. I further believe that this data would be interesting to include in the supplementary material of this study to serve as a quality control tool for future studies.

Thanks. For each dataset, we conducted two-sample two-sided t-tests. Two datasets (LIBD_151674 and mouse_brain) have $p\text{-value} \leq 0.05$ (not adjusted for multiple tests), where the estimated bleeding rates are higher in mitochondrial genes and long non-coding RNAs than the remaining genes. We have added these results to Supplementary Figure 6, with a note in the manuscript (lines 93-94). We have also added a paragraph to the Methods section detailing how the proportions and p-values were calculated.

Supplementary Figure 6: Boxplots of proportion of UMI counts in background spots for mitochondrial genes and long non-coding RNAs (mt_inc) and the remaining genes (not mt_inc) in 12 publicly available datasets as well as the chimeric experiments. Two datasets (LIBD_151674 and mouse_brain) show significant differences at $p\text{-value} \leq 0.05$ (and only mouse_brain if adjusting for multiple tests), where the estimated bleeding rates are higher in mitochondrial genes and long non-coding RNAs than the remaining genes. In general, there is no clear pattern where mitochondrial genes and long non-coding RNAs have higher or lower bleeding rates than the remaining genes.

3. Thank you for elaborating further on the motivation of this statement. While I agree with the authors that the diffusion of spots decreased the number of UMIs in the original spot (beneath the tissue), I consider the statement that these expression levels reduce the power of the entire experiment to be quite strong. Thus, if the authors want to stand by this comment I believe they need to show if UMI counts are in fact reduced significantly or rather have a large effect size in neighboring spots. One spot generally harbors a high number of UMIs (this is at least expected for highly expressed genes). Statistically, once sample sizes (UMIs) are very high, even small differences become significant, which might not necessarily be of the same biological significance and might therefore not have a meaningful effect on the interpretation of the observations.

Alternatively, the authors could simply remove this statement, so it reads: “While this possibly reduces expression levels at tissue spots, a bigger concern is spot swapping from one tissue spot to another, as this confounds downstream analyses.”

We have changed this sentence (lines 95-97) to read “While spot swapping from tissue to background reduces expression levels at affected spots, a bigger concern is spot swapping from one tissue spot to another, as this confounds downstream analyses.” We removed “possibly” from the suggested sentence, but added “affected”, since for spots affected by spot-swapping from tissue to background (i.e. bleeding of reads into the background), the effect is reduced expression.

Suggestion:

*** 10.I appreciate the author's effort to address the suggestion and agree with the authors that there is no clear correlation between permeabilization time and bleeding or distal rate in the analyses they present. I would ask the authors to consider including these findings in the supplementary data to first, show that their experiment is not influenced by the quality or permeabilization time of the tissue section and second, to advocate that the research community can use the SpotClean method to calculate bleeding and distal rates (using SpotClean) to potentially explain unexpected or unclear observations in their spatial experiments.**

Thanks. We have added these results to Supplementary Figure 7, with a note in the manuscript (lines 94-95).

Supplementary Figure 7: Relationship between bleeding rate estimated via SpotClean and permeabilization time (left) and between distal contamination rate estimated via SpotClean and permeabilization time (right). No clear patterns were observed between estimated bleeding rates and permeabilization times, or between estimated distal contamination rates and permeabilization times.

Reviewer #2 (Remarks to the Author):

the authors have addressed all my concerns. Congratulations for this nice work.

Thanks!